# Framing of grid cells within and beyond navigation boundaries

**Francesco Savelli[1]\*, JD Luck[1], James J Knierim[1,2]\***

[1]Zanvyl Krieger Mind/Brain Institute, Johns Hopkins University, Baltimore, United States; [2]Solomon H. Snyder Department of Neuroscience, Johns Hopkins University, Baltimore, United States

**Abstract** Grid cells represent an ideal candidate to investigate the allocentric determinants of the brain's cognitive map. Most studies of grid cells emphasized the roles of geometric boundaries within the navigational range of the animal. Behaviors such as novel route-taking between local environments indicate the presence of additional inputs from remote cues beyond the navigational borders. To investigate these influences, we recorded grid cells as rats explored an open-field platform in a room with salient, remote cues. The platform was rotated or translated relative to the room frame of reference. Although the local, geometric frame of reference often exerted the strongest control over the grids, the remote cues demonstrated a consistent, sometimes dominant, countervailing influence. Thus, grid cells are controlled by both local geometric boundaries and remote spatial cues, consistent with prior studies of hippocampal place cells and providing a rich representational repertoire to support complex navigational (and perhaps mnemonic) processes.

## Introduction

Different types of neural correlates of space are found in the hippocampal formation, including place cells, grid cells, boundary cells, and head direction cells (*O'Keefe and Dostrovsky, 1971*; *Taube et al., 1990a*; *Savelli et al., 2008*; *Solstad et al., 2008*; *Lever et al., 2009*). These cells signal the animal's position or direction relative to the external world; that is, they represent space allocentrically, similar to a 'you are here' mark on a geographic map. The allocentric property has implicated these cells as the neural substrate of a 'cognitive map' of the environment (*O'Keefe and Nadel, 1978*; but see *Bennett, 1996*; *Filimon, 2015*). Despite decades of intensive investigation, how the hippocampal circuits create this map is still not understood. More specifically, the question of what environmental cues—if any kind in particular—provide the allocentric reference frame of the internal map remains controversial.

Behavioral studies (*Cheng, 1986*; *Hamilton et al., 2007*; *Tommasi et al., 2012*) and electrophysiological studies of place cells (*O'Keefe and Conway, 1978*; *Knierim and Rao, 2003*; *Siegel et al., 2008*; *Samsonovich and McNaughton, 1997*; *O'Keefe and Burgess, 1996*) and head direction (HD) cells (*Taube et al., 1990b*; *Zugaro et al., 2001*) inferred two major allocentric determinants of the internal map's reference frame: distal (inaccessible) landmarks and the geometric configuration of the proximal (accessible) boundaries defining the animal's navigation range. Different studies often disagree on their relative influence (*Knierim and Hamilton, 2011*). Furthermore, the firing correlates of the two most extensively investigated cells (HD cells and place cells) can complicate the interpretation of these studies. HD cells provide an orientation/direction signal, but they do not provide a position signal, limiting experimental investigations to rotational manipulations and limiting the analysis to the directional frame. Place cells usually produce a single place field in a standard laboratory experiment, and therefore rotations and translations of the internal map can only be discerned unequivocally from the collective response of a population of them. But place cells can

\*For correspondence: fsavelli.
research@gmail.com (FS);
jknierim@jhu.edu (JJK)

**Competing interests:** The authors declare that no competing interests exist.

remap independently of each other (*Bostock et al., 1991*; *Colgin et al., 2008*), can turn on and off in a familiar environment or in response to cue manipulations (*Shapiro et al., 1997*; *Monaco et al., 2014*), and can be modulated by nonspatial aspects of the animal's experience (*Wood et al., 1999*, *2000*; *Frank et al., 2000*; *Moita et al., 2003*). In contrast, a grid cell fires in space according to a periodic, triangular pattern that simultaneously reveals its positional (by its phase) and directional (by its orientation) anchoring to the external world. Unlike place cells, grid cells do not undergo 'global remapping' (*Leutgeb et al., 2004*), but instead they are active and have similar spatial correlates in all environments (*Hafting et al., 2005*; *Fyhn et al., 2007*; *Marozzi et al., 2015*). Grid cells are thus ideal candidates to elucidate the allocentric nature of the internal map and its neural basis.

Grid cells are presumed to provide a universal metric for the animal's movements in space and for the cognitive map (*Moser et al., 2008*; but see *Krupic et al., 2016*). Studies to date have mostly concentrated on the role of the proximal navigation boundaries and their geometric configuration in the anchoring (*Savelli et al., 2008*; *Solstad et al., 2008*; *Stensola et al., 2012*, *2015*; *Krupic et al., 2015*), shaping (*Barry et al., 2007*; *Stensola et al., 2015*; *Krupic et al., 2015*), compartmentalization (*Derdikman et al., 2009*; *Carpenter et al., 2015*), and metrical correction (*Hardcastle et al., 2015*) of the grid pattern. Current theoretical proposals consequently reflect the prevailing experimental focus on these environmental features. For example, it has been suggested that grid cells are primarily concerned with representing geometric information about the local enclosure (*Krupic et al., 2016*), and emphasis has been placed on the role played by boundaries in stabilizing or differentiating grid maps (*Giocomo, 2015*; *Stensola and Moser, 2016*). In contrast, the influence of remote cues remains under-investigated. Although grids were shown to realign or rescale in a context- or novelty-dependent manner after all distal cues were changed (*Fyhn et al., 2007*; *Barry et al., 2007*), the cues were not subject to spatial manipulations in these experiments (but see *Neunuebel et al., 2013*; *Gupta et al., 2014* for experiments with circular tracks and T-mazes). Motivated by the well-known influence of remote inputs on non-metric neural correlates of space (*O'Keefe and Conway, 1978*; *Taube et al., 1990b*; *Knierim and Hamilton, 2011*), we investigated the relative influence of remote landmarks and local geometric cues on grid cells recorded in an open field.

## Results

To gauge the degree of control exerted on grid patterns by the distal landmarks vs. the geometry of the proximal boundaries, we dissociated the reference frames embodied by these two sets of cues. Visually prominent cues were affixed to the walls of the experimental room (355 × 280 cm, *Figure 1A,B*). We translated or rotated a square foraging platform (137 × 137 cm) between recording sessions, starting from a standard position (STD) in which the platform was parallel to the experimental room (*Figure 1A,C,E*). Two types of platforms were used with two separate sets of rats. One had walls (35.6 cm) and replaceable floor paper producing a luminance contrast between the walls and the floor (five rats, *Figure 1D*, left). The other had small lips (2.5 cm) around the perimeter and its floor was not replaceable (two rats, *Figure 1D*, right). Results from all 308 grid cells recorded over multiple days in the medial entorhinal cortex (MEC) and/or para-subiculum (*Figure 1F*) of the seven rats are described and included in statistical tests. These analyses provide a rich description of the reproducibility of the grid cell responses over multiple days of experimentation, but they also lead to complications regarding statistical analysis. To ameliorate concerns about the artificial inflation of statistical power by repeated sampling of the same units, the statistical tests were repeated after datasets were algorithmically pruned to minimize the possibility of resampling individual units ('thinned datasets', see Materials and methods for details). These tests are reported if they differed from the full dataset in failing to reach statistical significance at α = 0.05.

### Grid anchoring to both local and remote cues in rotation experiments

We first investigated rotations of the local reference frame relative to the global reference frame. Due to the 60° hexagonal symmetry of the grid pattern, we first considered responses to small rotational manipulations. Following a 20° clockwise (CW) platform rotation (ROT20), virtually all grids from all rats rotated by a similar amount (*Figure 2A,B*). However, a reliable influence of the room was still detectable, as the grids systematically under-rotated with respect to the platform rotation angle (20°) (mean rotation = 14° CW, 109 units pooled from five rats, p<0.0001; see

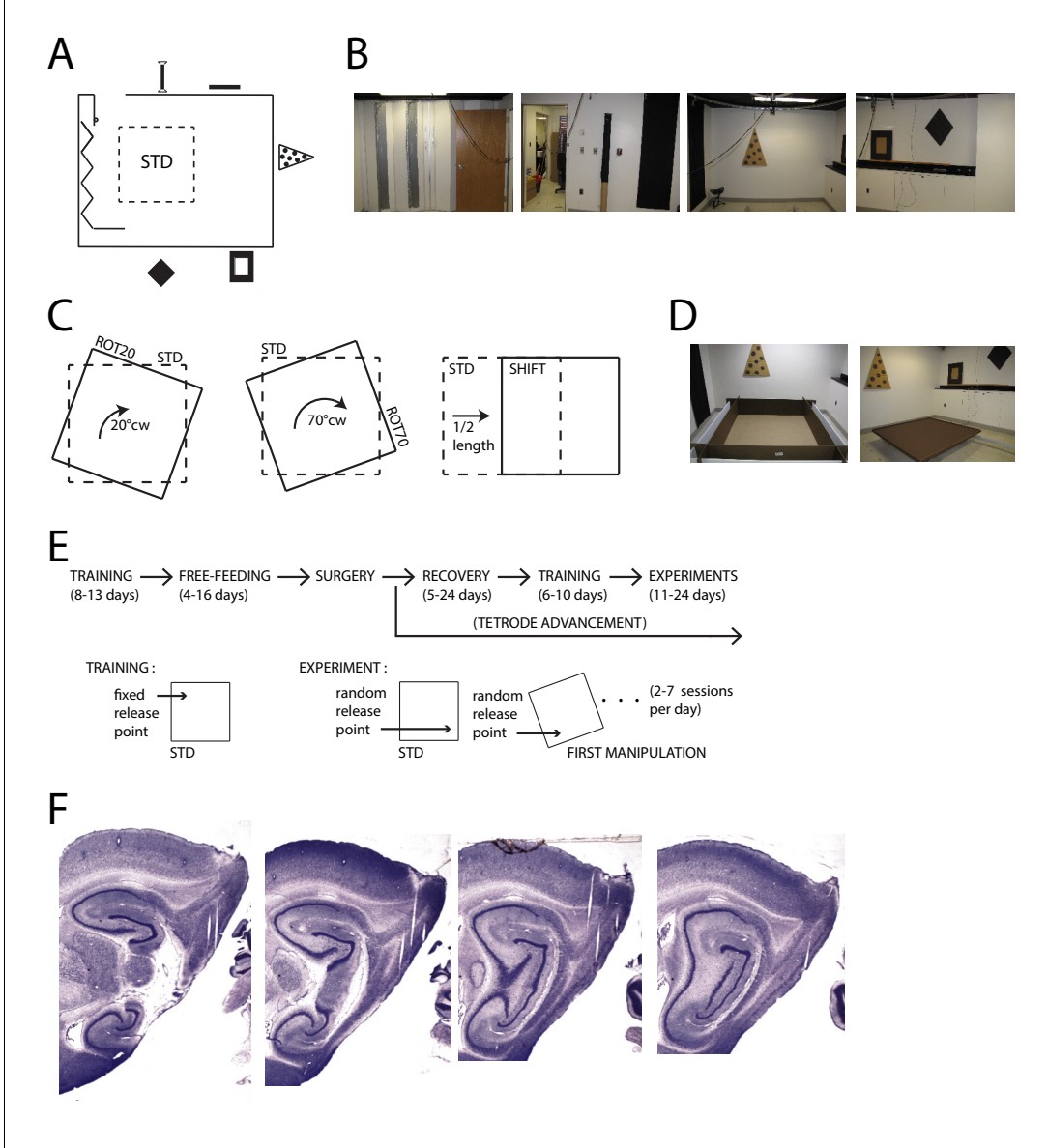

**Figure 1.** Apparatus and Recordings. (A) Schematic of the room and foraging platform in STD position, approximately to scale. Locations of wall cues are depicted around the room perimeter. (B) Panoramic set of photos portraying all the room walls. (C) Schematics of the type of experimental manipulations: 20° CW rotation (ROT20, left), 70° CW rotation (ROT70, center, geometrically equivalent to a 20° CCW rotation), and platform translation (SHIFT, right). (D) Pictures of the two types of platforms: with high walls (left) or small lips (right). (E) Timeline of pre/post-surgery training, subsequent experiments, and their difference in the protocol. During training only the STD configuration was used, and the rat was always released from the same quadrant of the platform (the same for all rats). Experiment days always started with an STD session and were followed by one or more manipulated sessions (occasionally STD was also repeated). The rat was released from a quadrant of the platform that was randomly determined at the beginning of each session (including STD) and removed from the platform at the end of it. A ROT70 is here illustrated as the first manipulated session of the day, but the sequence of manipulations typically varied from day to day in a semi-random fashion. (F) Representative sagittal histological sections with tetrode tracks from one rat (rat 377).

Materials and methods for description of how p values were calculated; also significant at p<0.0001 for each of the four rats with sufficient samples [n > 7] for a valid test). Similar under-rotations were observed with 30° CW platform rotations in one rat (*Figure 2—figure supplement 1*). In contrast to the grid rotations, the shift of grid phases in ROT20 was very small (mean ± S.D., 3.5 ± 4.0 cm, 5 ± 5% of grid period, *Figure 2C*).

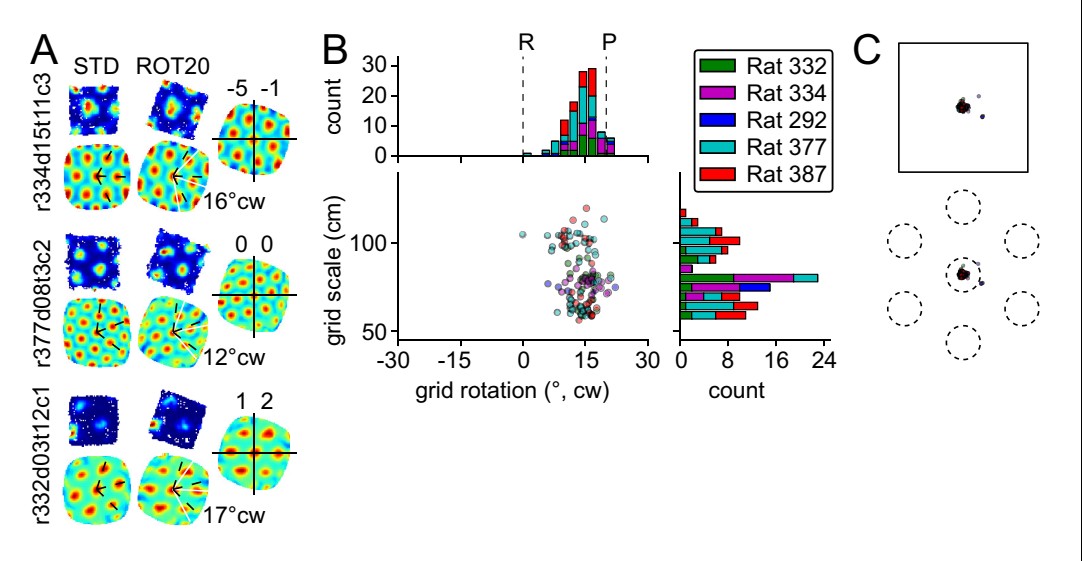

**Figure 2.** Response to platform rotation by 20° CW (ROT20). (A) Individual examples of rate maps for three units in the STD and ROT20 sessions. Top, rate maps calculated in the room reference frame. Bottom, autocorrelograms, from which the orientations of the three canonical grid axes (dashed black: STD, solid white: ROT20) were extracted and the grid rotation (mean angular difference between the two sets of axes, noted to the right of the autocorrelograms) was calculated. Right, rotation-adjusted crosscorrelogram between the two rate maps, computed after the STD rate map was rotated to equalize the orientation of its grid with that in ROT20. Center (0,0) of the crosscorrelogram is marked with cross-hair. Phase shift (noted above the crosscorrelogram in x,y cm) is the vector between the center of the crosscorrelogram and that of its closest correlation field. Each example shows that the grid did not rotate as much as the box, under-rotating by 4°, 8°, and 3°. In contrast, there was little phase offset between the two grid patterns. Unit ID is at the left of the figure (r: rat number; d: day of recording; t: tetrode number; c: cell number on that tetrode). (B) All grid rotations in ROT20. Dashed lines indicate the expected grid rotation for ideally room-controlled (R) or platform-controlled (P) grids. The grids on average rotated in the CW direction, with an undershoot of ~6°. Note that because of the 60° rotational symmetry of the grids, the abscissa represents a circular coordinate system, with the −30° value equal to the +30° value. Grids of different scale (ordinate) responded equivalently (see *Figure 8*). (C) Phase shifts for the grids in B, calculated on rotation-adjusted crosscorrelograms as in A. Top: Absolute phase shift magnitude. The amount of phase shift was very small compared to the size of the platform (superimposed shape of the platform is to scale). Bottom: Phase shift magnitude as a proportion of grid period. When the phase shift for each cell was plotted as a fraction of the grid period for that cell, the average shift is much smaller than the size of an individual grid vertex (superimposed grid is in an arbitrary orientation).

The following figure supplement is available for figure 2:

**Figure supplement 1.** Response to platform rotation by 30° CW (ROT30) in one rat.

Platform control of the grids could reflect either an influence of the local landmarks within the platform itself (e.g., uncontrolled cues such as odors or subtle markings) or the geometric structure of the platform boundaries. To disambiguate these alternatives, we rotated the platform 70° CW. If the grids were controlled by the platform itself, we would expect a 70° CW rotation with respect to the room, which would be measured as a 10° CW rotation due to the 60° symmetry of the grid. On the other hand, if the grids preserved the STD alignment relative to the platform's geometric structure, but rotated relative to the physical platform itself, we could observe a 20° CCW rotation with respect to the room, which is geometrically congruent to a 70° CW rotation (*Figure 1C*). (A third possible outcome is that the grid remains anchored to the room reference frame, yielding a 0° rotation.) In contrast to the uniform response to the ROT20 manipulation, grid rotations in response to the ROT70 manipulation formed a bimodal distribution (*Figure 3A*). One of the modes ('left mode', 50 units from three rats) was located at ~20° CCW (mean 18°), representing grids that were controlled by the platform's geometric structure (the 'geometric' reference frame), but rotated relative to the physical platform itself (test of mean angle vs. 10° CW, p<0.0001; also significant at p<0.0001 for each rat with n > 7). In one of the two rats that accounted for most of the data in this mode, rotations of the grid and the platform geometric frame matched (rat 332, mean rotation 21° CCW, test of mean angle vs. 20° CCW, p>0.05). In the other rat of the left mode, grids slightly under-rotated

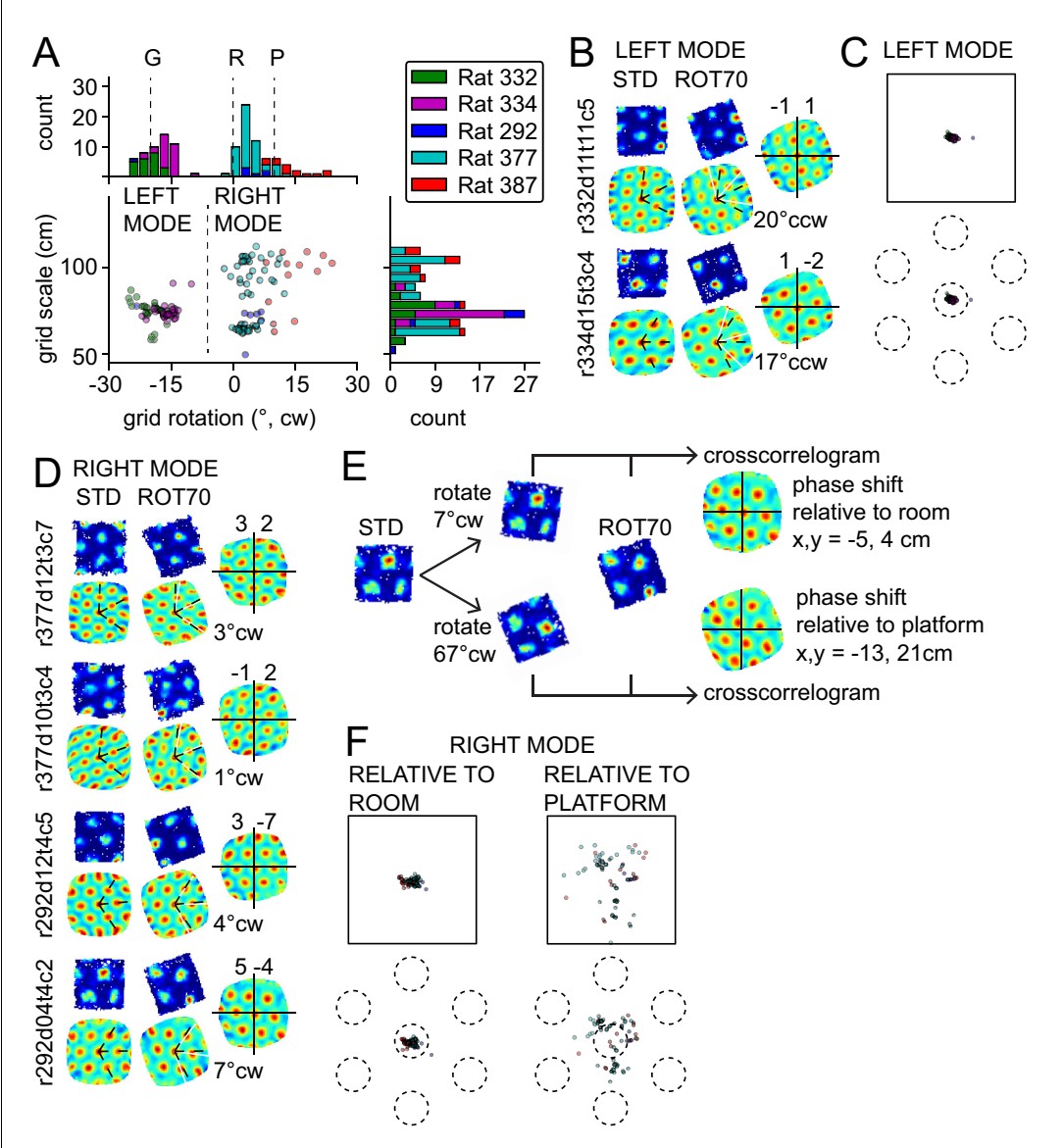

**Figure 3.** Bimodal response to platform rotation by 70° CW (ROT70). (**A**) A CW rotation of 70° allows the dissociation of the influences of the platform ('P' at 10° CW, corresponding to a 70° CW grid rotation because of the 60° radial symmetry of the grid), the room ('R' at 0°), and the geometric frame of the platform ('G' at 20° CCW, the minimum angle yielding a congruent square configuration for ROT70). This manipulation resulted in a bimodal grid-rotation response (abscissa). The left mode represents grids that were controlled primarily by the geometric frame of reference. The right mode represents grids that were controlled primarily by the room frame of reference (see illustration in *Figure 3—figure supplement 1*). Offsets of the modes from the predicted orientations indicate the counterweighing influence of a competing frame of reference in most rats. The 50 units in the left mode include 35 units that were recorded in the ROT20 manipulation (*Figure 2B*) on the same day. The 69 units in the right mode include 42 units that were recorded in the ROT20 manipulation on the same day. (**B**) Examples from the left (geometry-controlled) mode of the rotation distribution in **A**, illustrated as in *Figure 2A*. (**C**) Phase shifts in the grids of the left mode in **A**, illustrated as in *Figure 2C*. (**D**) Examples of rate maps from the right (room-controlled) mode of the rotation distribution in **A**. (**E**) Demonstration of phase shift calculation relative to room and platform reference frames for the last example in **D**. The grid rotates 7° CW in the room reference frame. This is equivalent to a 3° CCW rotational error relative to the physical platform's rotation of 70° CW (after the 60° rotational symmetry of the grid is subtracted from 70°). The phase shift relative to the platform is therefore computed from the crosscorrelogram between the STD rate map rotated by 70° CW +3° CCW = 67° CW and the ROT70 rate map. (**F**) Phase shifts calculated relative to the room and platform reference frames as in E for all grids in the right mode in **A**. Note how grid phase (i.e. the position of the grid) is poorly controlled by the platform, but tightly controlled by the room.

The following figure supplements are available for figure 3:

**Figure supplement 1.** Illustration of a room-anchored grid.

*Figure 3 continued on next page*

*Figure 3 continued*

**Figure supplement 2.** Response to platform rotation by 45° CW (ROT45) in one rat.

(rat 334, mean 16° CCW, p<0.0001), similar to ROT20. Also similar to ROT20, very little shift of grid phase was observed for all the grids in this mode (mean 3.6 cm, 5% of grid period, *Figure 3C*). The grids in the left mode, therefore, did not follow uncontrolled cues of the platform, but rather treated its orthogonal axes as interchangeable under the influence of the room cues.

In all of the responses described so far, the grids were primarily anchored to the platform's geometric boundaries, but the room cues broke the geometric symmetry of the square platform and also exerted a countervailing influence that tended to produce an under-rotation of the grid. By contrast, the other mode in *Figure 3A* ('right mode', mean 6° CW, 69 units from three rats) reflects grids that kept a stronger relationship with the room's distal landmarks/walls than with the platform's geometric boundaries (*Figure 3D*). The location of this mode is approximately consistent with the rotation expected of room-controlled grids (grid rotation = 0°, *Figure 3A* 'R'). In some cases, the local boundaries were almost completely ignored, as if the platform were a sampling aperture that unmasked a different region of the grid (*Figure 3—figure supplement 1*), capturing varying sets of vertices and/or varying fractions of the same firing fields (*Figure 3D*). However, the location of the right mode is also consistent with an under-rotation of grids following the full 70° CW platform rotation (grid rotation = 10° CW, *Figure 3A* 'P'). Taking the phase shifts of these grids into account resolves which reference frame dominated (*Figure 3E*, see also further explanation in Materials and methods). The phase shifts necessary to align each STD grid to its corresponding ROT70 grid were much larger when calculated relative to the physical platform compared to the room, both in absolute space and as a proportion of the grid period (phase shift magnitude, $29 \pm 12.9$ cm vs. $6 \pm 3.7$ cm, Wilcoxon signed-rank test, $W_{(69)} = 8$, $p < 7.5 \times 10^{-13}$; $34 \pm 11\%$ vs. $7 \pm 5\%$ proportion of grid period; *Figure 3F*). The phase shifts calculated based on a hypothesis that the grids randomly reoriented to any of the 4 sides of the platform were also larger than phase shifts based on the room-based reference frame (data not shown). These grids therefore strongly dissociated from both the physical platform (by phase shift) and its geometric structure (by rotation), and were instead controlled predominantly by the room reference frame. Grids dissociating from the proximal reference frame were also observed in one rat after 45° CW rotations (*Figure 3—figure supplement 2*). The difference of response expressed by the bimodal distribution of rotations in ROT70 (*Figure 3A*) is unlikely to result from the sampling of functionally differentiated neuronal networks (*Table 1*).

## Grid anchoring to both local and remote cues in translation experiments

In the translation manipulation (SHIFT, 119 units from seven rats), the platform was shifted by half its length along the room's longer axis (*Figure 1C*). The firing patterns expressed by grid and other spatial cells on the platform displayed a striking degree of similarity between STD and SHIFT (*Figure 4A–C*), suggesting that they remained anchored to the platform. While the grid phase shift can provide a quantitative indication of the grid displacement in SHIFT, by definition this measure cannot exceed the grid spatial period, which must be taken into account to interpret the shift of each grid. Hence we calculated (1) the observed STD-SHIFT phase shift relative to the platform and (2) the phase shift relative to the platform that is predicted by a room-bound expansion of the STD grid to the region of the room occupied by the platform in the SHIFT condition (see Materials and methods). The difference between (1) and (2) represents the phase shift relative to the room reference frame. The room-relative phase shifts were much larger than the platform-based phase shifts, both in absolute space and as a proportion of the grid period (phase shift magnitude, $14 \pm 10.1$ vs. $5 \pm 2.6$ cm, Wilcoxon signed-rank test, $W_{(119)} = 507$, $p < 4.6 \times 10^{-16}$; $18 \pm 11\%$ vs. $7 \pm 4\%$ proportion of grid period; *Figure 4D*, see also *Figure 5—figure supplements 1–7*), consistent with positional control exerted primarily by the platform. We also calculated direct correlations between the regions of the STD and SHIFT rate maps that are expected to overlap under the competing

**Table 1.** Anatomical distribution of the grid cells recorded in ROT70. All 119 units reported in *Figure 3A* and the tetrodes from which they were recorded are here counted by mode of the response, brain area, and rat. Note that the same brain areas are represented in both modes, sometimes in multiple rats and by multiple tetrodes in the same rat. The different response accounted by the two modes is therefore unlikely to depend on a functional differentiation across brain areas or within the same area. Differences in the animal's individual experiences and/or apparatus types appear more likely explanations. (MEC: medial entorhinal cortex; L2, L3, L2/3: respective layers of MEC; ParaS: parasubiculum.)

| Rot70 mode | Brain area | Rat id (APPARATUS) | UNITS # | TETRODES # |
|---|---|---|---|---|
| Left Mode | L2 | 332 (w/ walls) | 22 | 3 |
| | | 334 (w/ walls) | 2 | 1 |
| | L2/3 | 334 (w/ walls) | 7 | 1 |
| | MEC/ParaS | 334 (w/ walls) | 2 | 2 |
| | ParaS | 334 (w/ walls) | 16 | 2 |
| | ParaS? | 292 (w/ walls) | 1 | 1 |
| Right Mode | L2 | 377 (w/ lips) | 6 | 1 |
| | | 387 (w/ lips) | 10 | 2 |
| | L3 | 377 (w/ lips) | 2 | 1 |
| | MEC/ParaS | 377 (w/ lips) | 41 | 1 |
| | ParaS | 377 (w/ lips) | 1 | 1 |
| | | 387 (w/ lips) | 3 | 1 |
| | ParaS? | 292 (w/ lips) | 6 | 1 |

hypotheses that the grid was either room-bound or platform-bound (*Figure 4E*). The room-bound correlations were dramatically lower than the platform-bound correlations (*Figure 4E*, red vs. blue; Wilcoxon signed-rank test, $W_{(119)} = 68$, $p<1.5 \times 10^{-20}$).

Even in this manipulation—in which the platform exerted such strong control over the grids—the influence of the room was detected, consistent with analogous place cell studies (*Knierim and Rao, 2003*; *Siegel et al., 2008*). The correlations in *Figure 4E* further improved if the rate maps were first realigned according to the detected grid rotation and shift (blue vs. black). Relative to the platform frame of reference, grids generally shifted slightly in the direction opposite to the performed translation, thus 'lagging' behind the platform (*Figure 4F*, examples in *Figure 4A*). The component of the grid phase shift relative to the platform along this axis was significant (t-test against 0, $t_{118} = 10.85$, $p<1.8 \times 10^{-19}$; significant at $p<0.05$ in 5/7 rats [4/7 in thinned datasets]). This systematic under-translation, alongside the under-rotations observed in ROT20 and ROT70 (*Figures 2–3*), is compatible with a residual 'pull' from the distal room frame. Moreover, grids tended to perform a small rotation in the CCW direction (mean 2° CCW; *Figure 4G*; examples in *Figure 4B*). This small CCW re-orientation was significant at $p<0.01$ in 4 of the five rats with n > 7; in the fifth rat the reorientation occurred in the CW direction ($p<0.01$).

## Anchoring biases by manipulation type

The under-rotations and under-translations recurring in these experiments indicate that grids typically dissociated from the dominant reference frame, even if these dissociations were sometimes minor (*Figures 2B*, *3A* and *4F,G*). In a given rat, these dissociations and the changes in dominant reference frame tended to be manipulation-specific and reproducible over time. This rat- and manipulation-specific stability is illustrated with polar scatterplots of grid rotations relative to all reference frames (*Figure 5A*, *Figure 5—figure supplements 1–7*). Grid rotation was plotted as the polar coordinate and day of recording was plotted as the radial coordinate. Grid rotations often clustered by type of manipulation over the course of many days (manipulations are color-coded and rotations cluster by color). To quantify the reproducibility of the grid rotation response over days, mean unit

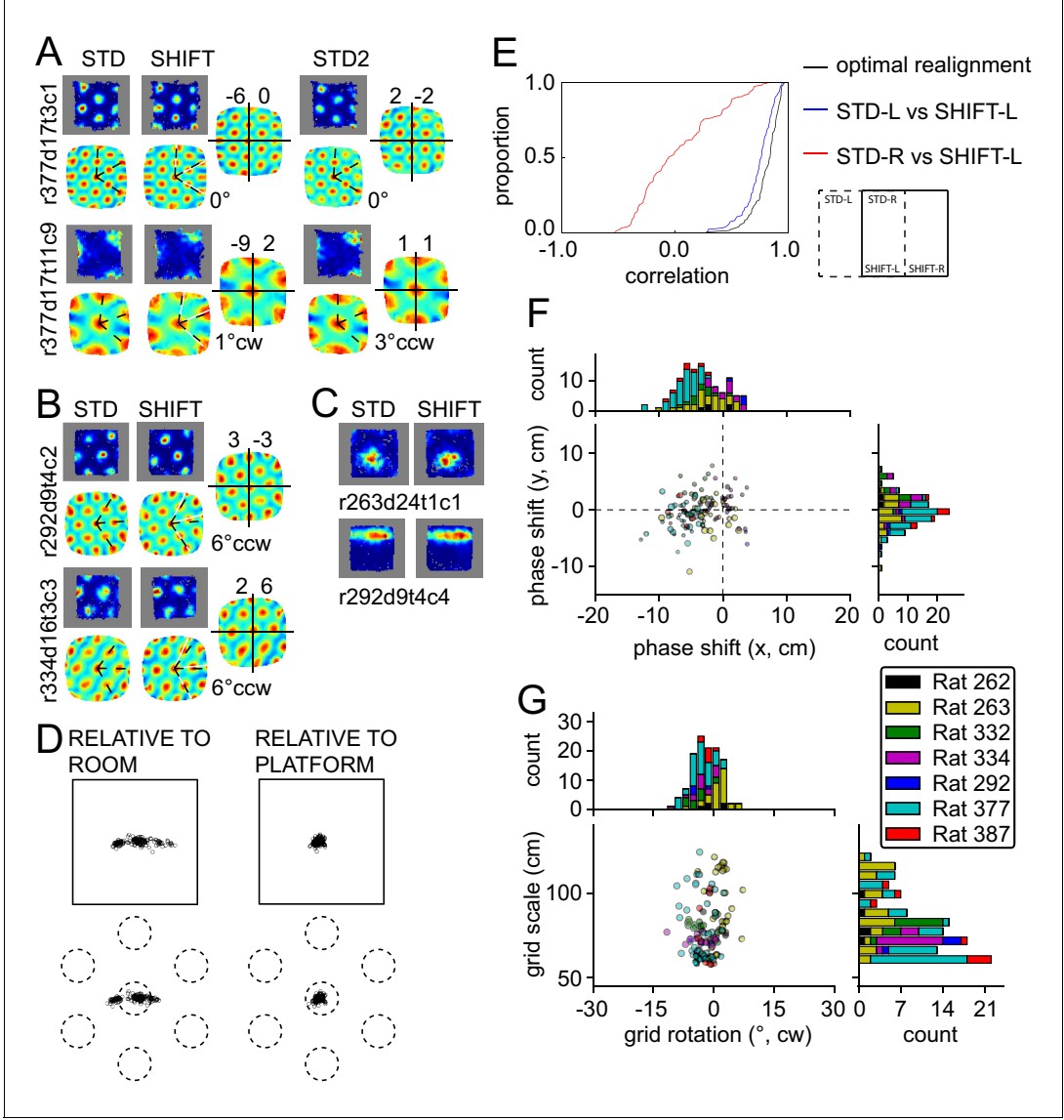

**Figure 4.** Response to platform translations (SHIFT). (**A–C**) Examples of rate maps illustrated as in *Figure 2A*. Grey areas outline identical subregions of the view frames of a camera positioned over the standard position of the platform and a separate camera positioned over the shifted position and aligned precisely with the first camera. (**A**) Examples of grids that shifted with the platform, with a small under-translation. (**B**) Examples of grids that shifted with the platform but that rotated slightly CCW. (**C**) Examples of additional spatial cells that shifted their firing fields with the platform. (**D**) Grid phase shifts relative to the platform or room, illustrated as in *Figures 2C* and *3F*. The 119 units include 77 that were recorded in the ROT20 (*Figure 2B*) and/or ROT70 (*Figure 3A*) manipulations on the same day. Although overall the phase shifts relative to the platform were much smaller than shifts relative to the room, note that a proportion of the grids showed a very small phase shift relative to the room as well. These are grids with spatial periods that were similar to the magnitude of the shift itself. (**E**) Cumulative distributions of correlations of STD vs SHIFT rate maps from different sections of the platform. In the SHIFT condition, the left half of the platform (SHIFT-L) offers a strong, direct test of the competing hypotheses. If the grids were controlled strongly by the platform, then the SHIFT-L rate maps should be highly correlated with the STD-L rate maps. In contrast, if the grids were controlled strongly by the room, then the SHIFT-L rate maps should be highly correlated with the STD-R rate maps. The STD-L vs. SHIFT-L correlations (blue) are much higher than the STD-R vs. SHIFT-L correlations (red), demonstrating conclusively that the grids are primarily controlled by the platform frame of reference. Nonetheless, after the grids are realigned to correct for any rotational or phase adjustments, the correlations are even higher (black), demonstrating that the room still exerts a measurable influence on the grids. (**F**) Individual grid phase shifts in the platform frame of reference, showing a small but consistent translational offset in SHIFT, in the direction opposite to the direction of platform shift in most animals (same data as in *Figure 4D*). Dot size is proportional to the grid scale measured in the STD session. (**G**) Individual grid rotations showing a minor CCW rotation in SHIFT in most animals.

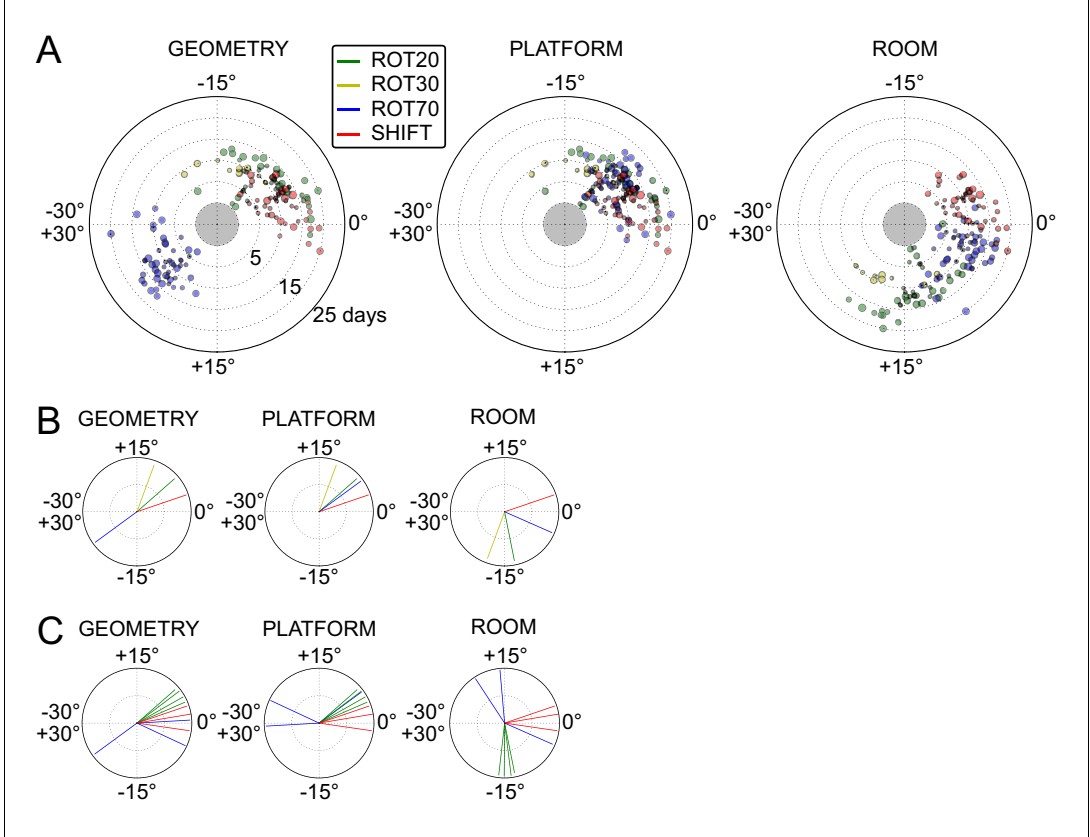

**Figure 5.** Systematic biases in frame prevalence and anchoring by rat and manipulation. (**A**) Rotation of each grid recorded from rat 377, organized by manipulation type (color), relative to the platform geometric reference frame (left), actual platform (center), and room (right). Perfect control by a reference frame is indicated by a 0° angle, whereas positive angles correspond to a CW rotational dissociation from the frame (i.e. the direction of the experimental rotations of the platforms). The radial position indicates the number of recording days since the rat's first experience of a non-STD session (external edge of the central gray circle). Dot size is proportional to the scale of the grid measured in STD. Note the general consistency of responses across days and the segregation of response angles by manipulation type (color). These plots ignore the phase component of the grid response, which dramatically reveals additional separation only in some animals/manipulations. Scatterplots for all seven rats are given in figure *Figure 5—figure supplements 1–7*, together with scatter plots for phase shift data. (**B**) Mean unit vectors of the rotation distributions calculated for each manipulation (color) from the data in **A**. Mean vector lengths are measures of circular variance. All mean vectors are close to 1 (maximum), consistent with the tight distributions of angles for each manipulation type. (**C**) Mean vectors from all rats and manipulations as in **B**, restricted to the rat-manipulation datasets that comprised at least seven distinct recoding days. As in (**B**), all mean vectors are close to the maximum value of 1, showing that the individual rats' responses to each manipulation were reproducible and stable across multiple days of recordings. See figure supplements 1-7 for more details for each rat.

The following figure supplements are available for figure 5:

**Figure supplement 1.** All grid rotations and phase shifts from rat 387.

**Figure supplement 2.** All grid rotations and phase shifts from rat 377.

**Figure supplement 3.** All grid rotations and phase shifts from rat 334.

**Figure supplement 4.** All grid rotations and phase shifts from rat 332.

**Figure supplement 5.** All grid rotations and phase shifts from rat 292.

**Figure supplement 6.** All grid rotations and phase shifts from rat 263.

**Figure supplement 7.** All grid rotations and phase shifts from rat 262.

vectors (*Figure 5B*) were calculated for each rat-manipulation coupling comprising >6 recording days (not necessarily contiguous), yielding 10 rat-manipulation couplings from five rats (*Figure 5C*). A random distribution of angles would result in a mean vector near 0, whereas perfect clustering would result in a mean vector of 1. In all 10 couplings regardless of reference frame, the rotations were significantly non-uniform (Rao test for circular uniformity, all couplings significant at p<0.001) and significantly different from 0° (all 10 couplings x three reference frames significant at p<0.05), indicating imperfect control by any single reference frame. The mean vector length in each coupling was very high (>0.92), demonstrating that the responses to each manipulation type were highly consistent over one week or longer.

Within individual rats, did different manipulations produce distinctive grid responses? To address this question, we computed signed differences of individual grid cells' orientations between manipulations recorded on the same day (i.e., direct estimates of the relative grid rotations across any two manipulations without recourse to STD as a shared reference). We grouped these differences by rat, reference frame used to compute the difference (geometry, platform, or room), and paired type of the manipulations involved in the difference (e.g., 'ROT20-ROT70', 'ROT20-SHIFT', etc.). Of the 36 such groups from four rats that met criteria for circular statistical testing (n > 7), 31 groups (each comprising 3–13 distinct recording days) from all four rats were found to be significantly different from 0° (all 31 groups significant at p<0.05; [in the thinned datasets only 6 groups from two rats met the n > 7 requirement and were all found significant at p<0.01]), indicating that relative rotations across identified manipulations were generally systematic with respect to all three reference frames. High angular concentration was measured in these groups (mean vector length >0.88 in all groups), further indicating that grid reorientations across the same manipulations were precisely reproduced over multiple days.

Taken together, these analyses indicated that grid angular drift from all three reference frames tended to be manipulation-specific and precisely controlled over time.

## Neural correlates of the local VS. remote cue conflict

Beyond the typical under-rotations and under-translations described above, idiosyncratic (but consistent within individual rats) neural correlates of the conflict between reference frames were observed.

In one rat (rat 387, platform with lips), the equilateral-triangular grid pattern was often selectively disrupted in ROT70, even though the cell continued to fire in discrete, multiple fields (*Figure 6A*, *Figure 6—figure supplement 1*). We considered all the units that were successfully recorded from this rat in ROT70 and at least another non-ROT70 session in a given day, if they passed the gridness test (see Materials and methods) in at least one of these sessions. Whereas 19/43 (44%) of the ROT70 sessions passed the gridness test, a larger proportion did so in all other sessions (84/131, 64%, $\chi^2$ test for proportions with Yates correction for continuity: $\chi^2(1)=5.3$, p<0.021). The selective loss of regular grid structure was apparent from the first day ROT70 was experienced by the rat and continued up to the last day in which grid cell recordings were available for ROT70, 17 days later (*Figure 6A*, *Figure 6—figure supplement 1*). The loss of gridness was not caused by a gradual, within-session drift of an otherwise well-formed grid: correlation of rate maps obtained from the first and second halves of each session were high in ROT70 and not different than the other sessions (median Pearson correlation 0.66, n = 43 vs. 0.68, n = 131, Mann-Whitney test, U = 2649, p>0.5). The rate maps that passed the gridness test in ROT70 had more elliptical distortion (*Stensola et al., 2012*) than those passing the gridness test in any other manipulation type for this rat (median elliptical index 1.2, n = 19 vs. 1.1, n = 93, Mann-Whitney test, U = 1352, p<0.0002). (These data were included in *Figure 3* in the same way as for other sessions and rats; examples are in *Figure 6—figure supplement 1B*.) Thus, in this rat, a strong influence of the room frame of reference was revealed by a striking, repeatable disruption of the periodicity of the grid pattern when the platform frame of reference was placed in conflict with the room frame of reference.

In another rat (rat 292, platform with walls), STD sessions repeated at the end of the day (STD2) consistently displayed a different alignment of the grid compared to the first STD sessions of the same day (*Figure 6B* (i)). This realignment was not random; rather, STD2 invariably reproduced the grid-platform alignment that was displayed in a previous, noncontiguous rotation manipulation (*Figure 6B* (i)). This phenomenon was observed every time a STD2 session was performed in this rat, spanning 10 experiment days across which the daily manipulation sequences varied (*Figure 6—figure supplement 2*). A few observations added to the surprising character of this phenomenon. First,

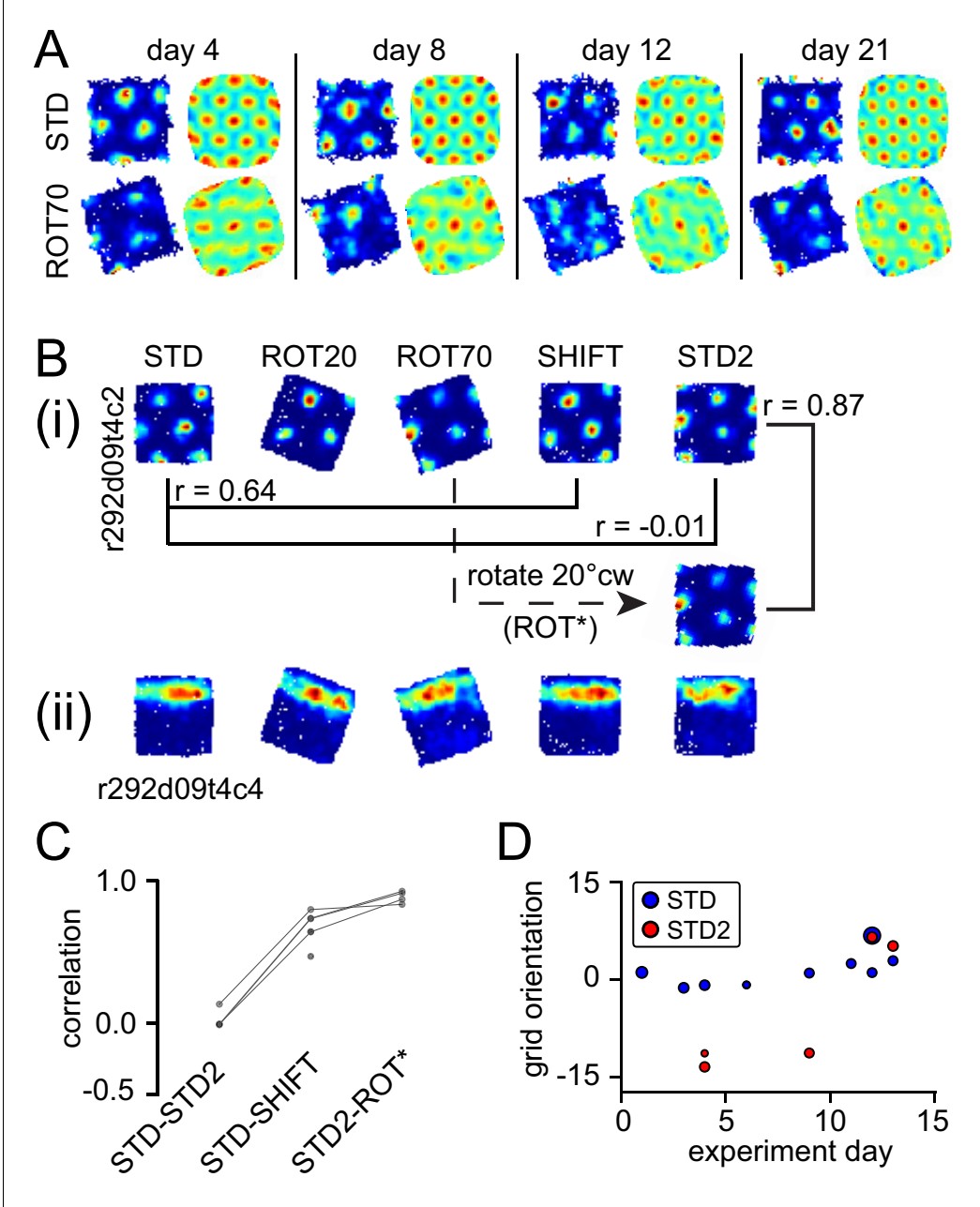

**Figure 6.** Neural correlates of cue conflict that are idiosyncratic but consistent within individual rats. (**A**) Grid degradation. Rate maps and autocorrelograms of 4 grid cells recorded from four different tetrodes in rat 387 in both STD and ROT70 on four different days (including the first and last days in which grid cells were obtained in ROT70). The equilateral triangular structure, but not the multi-field nature, of the grid cells was dramatically reduced or entirely lost in ROT70. Rate maps for all sessions in which these cells were recorded and additional examples are given in figure supplement 1. (**B**) Grid realignment and dissociation from boundary cells. Rate maps for two units and all sessions of one experiment day, from rat 292. After varying amounts of realignment in the ROT20 and ROT70 conditions (**i**), the grid reverted back to its standard alignment relative to the platform boundaries in the SHIFT condition and shifted along with the platform (with a minor CCW rotation, as described in *Figure 4B and G*). However, in STD2, the grid did not maintain this standard alignment. Instead, the grid reverted to the same alignment relative to the platform boundaries that it had adopted in the ROT70 session. Pearson correlations between relevant maps quantify the poor overlap between STD and STD2 and the strong overlap between ROT70 and STD2 (after the ROT70 was rotated 20° CW). A simultaneously recorded boundary cell (**ii**) always fired along the upper platform boundary, dissociating from the realignment of the grid in ROT20, ROT70,

*Figure 6 continued on next page*

*Figure 6 continued*

and STD2 (re-illustrated as first example in *Figure 7A*). (Additional examples are given in *Figure 6—figure supplement 2* and *Figure 7A*.) (C) Pearson correlations between rate maps from STD vs. STD2 (n = 4), STD vs. SHIFT (n = 6), and STD vs. the manipulation found to induce the new alignment in STD2 (n = 4), in a given day, for all grid cells and days in rat 292. Values referring to the same unit and day are linked. (D) Grid orientations in the standard sessions sampled over the course of 13 experiment days. Dot size is proportional to grid scale. Grid orientations in STD sessions (blue) are consistent across days in spite of the change of orientation experienced in STD2 sessions (red).

The following figure supplements are available for figure 6:

**Figure supplement 1.** Selective loss of hexagonal grid structure in rat 387.

**Figure supplement 2.** Bistable anchoring of grids across reference frames and experimental sessions in rat 292.

intervening SHIFT sessions were unaffected and continued to approximately replicate the STD, platform-bound alignment (*Figure 6C*, *Figure 6—figure supplement 2*) as described above for all rats (see *Figure 4*). Second, the STD2 alignment did not carry over to the STD sessions of the next day (*Figure 6D*, *Figure 6—figure supplement 2*); rather, the grids reset to their standard alignment at the start of each recording day. Third, some boundary cells did not change their firing patterns between STD and STD2 when the simultaneously recorded grid cells realigned. Instead, these cells continued to fire along the same geometric boundary in both sessions (*Figure 6B* (ii)), thus dissociating from the grids in these and other sessions (*Figure 7A*, *Figure 6—figure supplement 2*). These observations show that a new grid-platform anchoring configuration, originally elicited by a conflict between the platform and room reference frames, can be reactivated later in the absence of such conflict, without irreversibly overwriting the more familiar (STD) configuration.

Examples of divergence between grid and boundary representations were also observed in a second rat (rat 377, platform with lips) when the grid strongly dissociated from the platform reference frame (the 'right mode' of *Figure 3A,D*). In this situation some boundary cells kept firing along the same geometric boundary (i.e., the boundary representation tracked the proximal geometric reference frame, unlike the grid representation) (*Figure 7B*, *Figure 7—figure supplement 1*). Firing fields of other boundary cells with a more ambiguous response are included in *Figure 7—figure supplement 1*. In a third rat (rat 387), the possibility of divergence of the two representations in the same conditions could only be suspected based on the recordings of a cell tracking the same geometric boundary in ROT70 and the general behavior of grid cells recorded at different times (*Figure 6—figure supplement 1C*).

To mitigate concerns that these idiosyncratic responses, as well as the previously described minor and major dissociations from the platform reference frame, resulted from spatially unstable grids, we extended the analysis employed above for rat 387 to the recording sessions obtained from the other rats. For each session lasting longer than 30 min we computed the Pearson correlation between the rate maps obtained from the first and second half of the session (191 rate maps in STD-type sessions and 349 rate maps in manipulated sessions from four rats). These correlations were found to be high in both STD (median correlation >0.68 in each rat) and manipulated sessions (median correlation >0.69 in each rat), suggesting that there was no major intra-session grid drift or change of anchoring within a session (correlations in STD were not different from those in manipulated sessions: Mann-Whitney test, U = 32572, p>0.48, all rats pooled together).

## Grid coordination within and between scales

In order to determine if the population of grid cells maintained a coherent response to the manipulations performed in this experiment, we quantified the spatial coupling of simultaneously recorded grid cells, both within and across grid scales. We considered all the possible pairings of grids that had quantitatively distinct spatial firing patterns (Pearson correlation of STD rate maps < 0.5; see Materials and methods) in a given session. We measured their coupling by (1) the difference of the two grid rotations elicited by the manipulation performed in that session, and (2) a joint correlation measuring how well the STD rate maps match the corresponding non-STD rate maps after they are

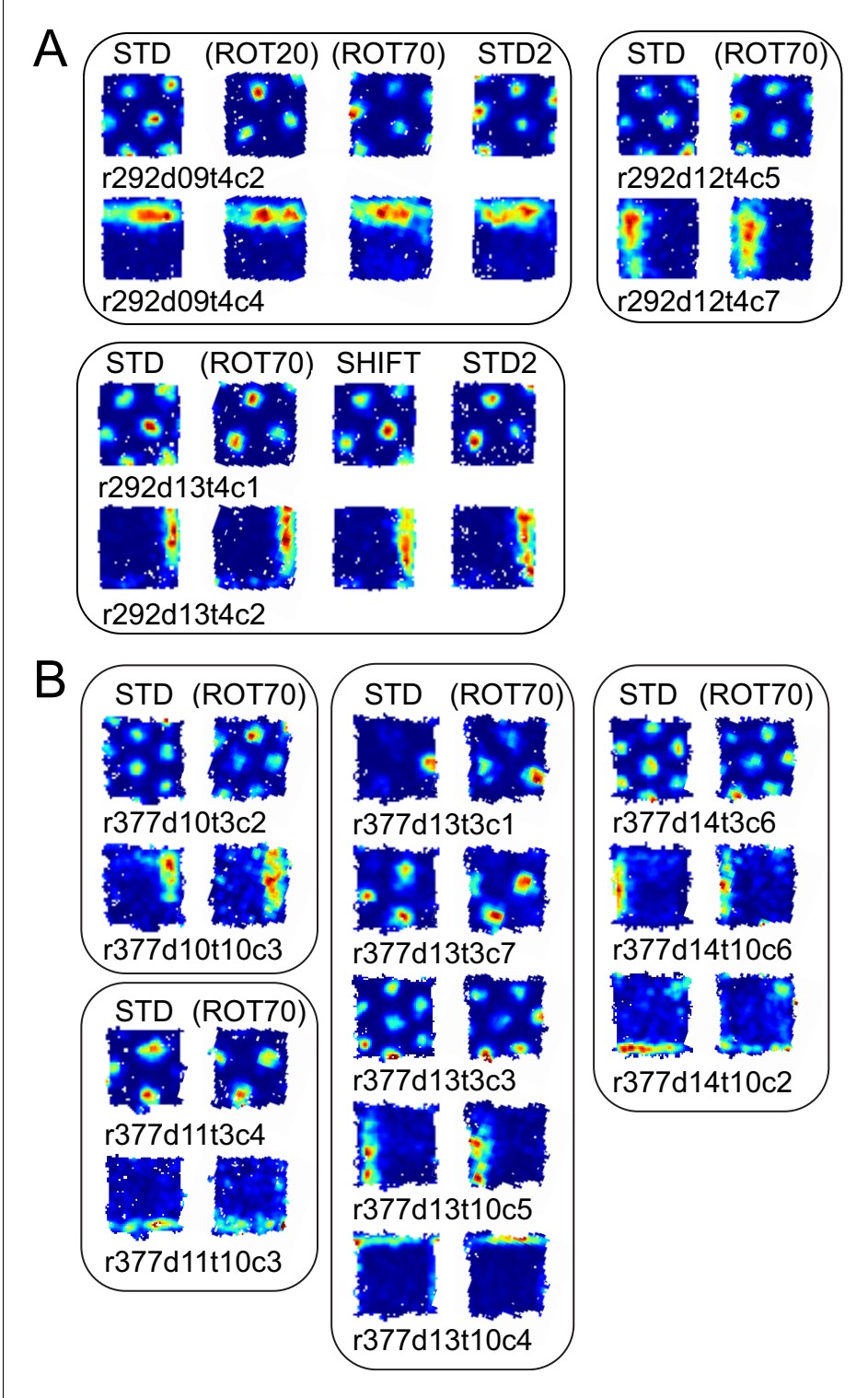

**Figure 7.** More examples of grid and boundary cells recorded simultaneously from rats 292 (**A**) and 377 (**B**). Each simultaneously recorded set is in a separate bounding box. The rate maps for sessions labeled in parentheses were rotated to aid visual comparison of the changing firing patterns of grid cells vs. the repeating patterns of boundary cells. Rate maps for all sessions in which these cells were recorded are given in *Figure 6*, *Figure 6—figure supplement 2* (rat 292) and *Figure 7—figure supplement 1* (rat 377).

*Figure 7 continued on next page*

*Figure 7 continued*

The following figure supplement is available for figure 7:

**Figure supplement 1.** Examples of grid and boundary cells recorded simultaneously from rat 377.

rotated and translated rigidly together by their average rotation and translation (see Materials and methods).

Of the 476 pairs from six rats, 81% displayed a rotation difference <5° and 88% had joint correlation >0.7 (*Figure 8A*), indicating high levels of geometric coordination. Visual inspection of rate maps also confirmed the phase/orientation coordination of grid cell populations simultaneously recorded (*Figure 8—figure supplement 1*, see also *Figure 6—figure supplement 2* and *Figure 7—figure supplement 1*). Because of the discrete representation of spatial scale in the grid system (*Barry et al., 2007*; *Stensola et al., 2012*), grid pairs segregated into three clusters according to their grid scale ratio (*Figure 8A*, SR1-3: 336, 135, and five pairs, respectively). In most pairs the two

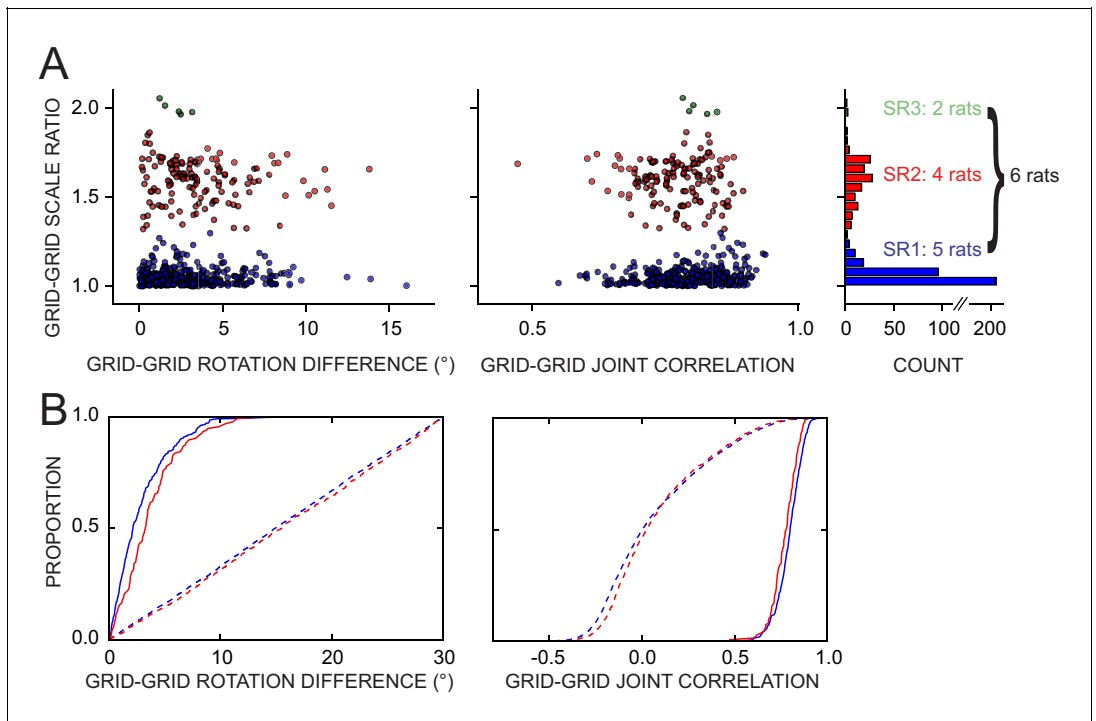

**Figure 8.** Response of simultaneously recorded grid cells. (A) Difference in rotations and joint correlations for pairs of grid cells from six rats in all manipulations, by the pair's grid scale ratio (SR): blue SR ≤ 1.3 (SR1); red: 1.3 < SR ≤ 1.9 (SR2); green: SR > 1.9 (SR3). Note that the scale-ratio clusters do not correspond to the absolute scales; rather, a scale ratio ~1.0 (blue dots) indicates two grids of the same scale (regardless of the absolute scale size) and a scale ratio ~1.6 indicates two grids from presumably adjacent modules (as defined by *Stensola et al., 2012*). The outliers in each scatterplot (rotation difference >10° on left, joint correlation <0.5 on right) were caused by six large-scale grids that participated in 11 pairs. Visual inspection of the rate maps of these six grids determined that poor spatial sampling of the peripheral vertices made these measurements ambiguous. (B) Cumulative density distributions for the SR1 and SR2 data in **A** (solid lines). For both measures, the two distributions were only slightly different from each other, indicating that grids across scales (SR2) were almost as coherent with each other as grids of the same scale (SR1). Dashed lines represent the control distributions generated by assigning a random phase and orientation response to each grid.

The following figure supplements are available for figure 8:

**Figure supplement 1.** Simultaneously recorded grid cells from one day during novel and familiar manipulations.

**Figure supplement 2.** Weak or no relationship between grid coordination and (A) magnitude of dissociation of grids from any reference frame and (B) amount of experience.

grid cells were anatomically adjacent as they were recorded from the same tetrode, but in many other cases they were recorded on different tetrodes (SR1-3: 119, 48, and all five pairs, respectively). In some of the latter cases the recording sites were ascertained to be in different layers of MEC or in MEC and parasubiculum (46 in SR1 and 9 in SR2 by a highly conservative histological evaluation). The distributions of rotation difference and joint correlation for SR1 vs SR2 diverged minimally (*Figure 8B*, blue vs. red solid curves; difference of medians for the rotation difference measure = 1.1°, Mann-Whitney test, U = 18012, p<0.0005 [lost significance in thinned datasets n1 = 62, n2 = 42, p>0.5]; difference of medians for the joint correlation = 0.02, U = 27202, p<0.0008), whereas very large differences were found between these distributions and corresponding control distributions obtained by randomly perturbing the grids' orientation and phase (*Figure 8B*, solid vs. dashed curves; difference of medians for the rotation difference measure, blue = 12.8° and red = 12.2°; difference of medians for the joint correlation, blue = 0.8 and red = 0.75). We then asked if grid coordination was influenced by the grid angular drift from any reference frame or by the animal's accumulated experience with the manipulation. We found only non-significant or very small correlations between grid coordination and these variables (*Figure 8—figure supplement 2*).

## Discussion

The present experiments demonstrate that the world lying beyond current navigation boundaries contributes to determining the allocentric reference frame of two-dimensional grid maps. While platform geometry usually exerted a dominant—but rarely absolute—influence on the grids, in some cases the grid map appeared embedded in the metric structure of the room and dissociated from the platform (*Figure 3D*, *Figure 3—figure supplement 1*). Neural correlates of boundaries could also decouple from the grids, reflecting further the ability of the grid map to dissociate geometrically from real-world boundaries.

This ability has fundamental implications. Grid cell properties can be reconciled or contrasted with those of place cells, which have long been known to respond to both extra- and intra-maze inputs (e.g, *Kelemen and Fenton, 2010*; see *Knierim and Hamilton, 2011* for extensive review). Early place cell studies emphasized the importance of distal cues (*O'Keefe and Conway, 1978*), but later studies amply documented the importance of local cues (*Shapiro et al., 1997*; *Knierim, 2002*; *Knierim and Rao, 2003*; *Lee et al., 2004*; *Renaudineau et al., 2007*; *Siegel et al., 2008*; *Kelemen and Fenton, 2010*), including geometric ones (*O'Keefe and Burgess, 1996*). Our observations motivate a similar revision/expansion, in reverse, of the early grid cell studies focusing almost exclusively on the influence of local, geometric cues (*Barry et al., 2007*; *Savelli et al., 2008*; *Derdikman et al., 2009*; *Stensola et al., 2012*; *Carpenter et al., 2015*; *Stensola et al., 2015*; *Krupic et al., 2015*). In our experiments differences in the relative dominance of distal and local cues appeared to depend on individual differences between rats and/or the type of apparatus, rather than on the anatomical regions where the grid cells were recorded (*Table 1*, *Figure 8*). However, the statistical validation of this impression would require data from many more rats than considered in our study.

Moreover, a grid cell system subject to distal control is ideally suited to fit Tolman's original inspiration for the cognitive map as a 'comprehensive-map' (*Tolman, 1948*, p. 193). Rats' ability to take novel shortcuts through previously inaccessible regions led Tolman to hypothesize the 'building up in the nervous system' of such allocentric maps (*Tolman, 1948*, p. 193). Grid cell-based path planning is theoretically possible (e.g, *Kubie and Fenton, 2012*; *Erdem and Hasselmo, 2012*; *Bush et al., 2015*) and robust to grid distortions such as shearing and boundary-induced perturbations (*Stemmler et al., 2015*). Planning a path between two familiar regions through territory that was previously blocked off, therefore, could in principle be accomplished via grid cells controlled by a distal reference frame comprising both regions. In this sense, interindividual variability in the shortcut errors (*Tolman, 1948*, see also *Grieves and Dudchenko, 2013*) is consistent with our observation that the grid cell map can be, but is not always, controlled by the distal laboratory reference frame, possibly with an associated angular error. In this putative framework, a remotely anchored grid map provides an implicit metric of currently inaccessible regions of the environment, which can later facilitate the pursuit of unanticipated navigation goals and opportunities. This framework differs from the hypothesis that the grid cell system is primarily dedicated to encoding the geometry of

local enclosures (*Krupic et al., 2015*, *2016*), and its further investigation requires laboratory environments that provide effective remote anchoring options for the animal's internal map, such as the apparatus we described.

The response of grid cells to the conflicting reference frames was diverse in frame preference, extent of under/over rotation/translation, and, in one rat, grid regularity. These differences were generally manipulation-specific and consistent over many recording days. Thus, they do not reflect repeated errors of a grid stabilization process that reacts haphazardly to the conflicting cues each time they are experienced. Rather, these repeatable phenomena possibly reflect an underlying discriminating process of the type that is hypothesized to produce different place cell maps and context-dependent memories through global remapping of place cell's firing locations (*Nadel et al., 1985*; *Anderson et al., 2006*; *Colgin et al., 2008*; *Wang et al., 2012*; *Fyhn et al., 2007*; *Stensola et al., 2015*; *Stensola and Moser, 2016*). Previous observations inspiring this hypothesis include the findings that global remapping or contextual changes are accompanied by grid realignment or expansion relative to a local enclosure (*Fyhn et al., 2007*; *Barry et al., 2012*; *Marozzi et al., 2015*) and that global remapping can be elicited through the partial inactivation of MEC (*Miao et al., 2015*; *Rueckemann et al., 2016*). In our experiments, the rats may have come to recognize and memorize each manipulation type as a separate experience or spatial context, which was consistently rendered by a reproducible change of dominant reference frame or (more often) by a distinct anchoring of the grid map to the same dominant reference frame. A more extreme example was noticed in one rat in which the regularity of the grid itself was dramatically disrupted for weeks in one type of manipulation only (ROT70). This highly selective grid disruption was triggered by the conflict between proximal and distal cues in the absence of any structural alteration of the platform. Thus the grid disruption cannot be attributed to tensions induced by the platform frame alone (*Barry et al., 2007*; *Derdikman et al., 2009*; *Stensola et al., 2012*; *Krupic et al., 2015*) but probably to the grid system's attempt to reconcile proximal and distal inputs. It is perhaps surprising that this response (as well as other seemingly 'dysfunctional' adaptations that we observed, such as under-rotations/translations) appeared to be consolidated instead of being corrected over the course of weeks of experience, similar to previous observations of grid distortions (*Derdikman et al., 2009*; *Stensola et al., 2012*, *2015*; *Krupic et al., 2015*). In fact, hypothesized functions of the grid system may be preserved even in the presence of grid distortions (*Stemmler et al., 2015*; *Carpenter and Barry, 2016*), and if small geometric idiosyncrasies of the inputs disproportionately contribute to triggering place-cell global remapping—as entailed by theoretical models relating grid cells to place cells (*Savelli and Knierim, 2010*; *Monaco et al., 2011*)—then their consolidation could well prove necessary for consistently recalling the correct place-cell map. A seeming counter-example to the involvement of grid framing in proper context recall was instead noticed in one rat in which different grid maps were produced in identical cue configurations at the beginning and the end of the experimental day (STD vs. STD2, *Figure 6B*, *Figure 6—figure supplement 2*). But even in this case a short-term memory process appeared at play, since we anecdotally found the later grid map to reflect the recent history of experimental interventions (*Gupta et al., 2014*), which could be regarded as a form of contextual discrimination.

In principle, the correction of path-integration errors required for stabilizing the grid could rely primarily on local boundaries (*Giocomo, 2015*; *Hardcastle et al., 2015*) even when room influence alters the geometric relationship between these boundaries and the grid, if the system has already learned the resulting grid-platform alignment for each manipulation. However, because the manipulation-specific grid realignments observed in our experiments were not randomly generated, stabilizing cues other than local boundaries seem necessary at least during the first experience of a manipulation before this learning occurs. In our apparatus, the distal visual landmarks on the room walls, or the room walls/boundaries themselves, were probably utilized to stabilize the grid in addition to local boundaries. Information about distal walls and landmarks may reach grid cells via boundary/landmark vector cells (*McNaughton et al., 1995*; *Hartley et al., 2000*; *Lever et al., 2009*; *Deshmukh and Knierim, 2013*) (*Figure 3—figure supplement 1*). Alternatively, the grids might have relied on room cues to reset their orientation via the HD cell system (*Winter et al., 2015*; *Zugaro et al., 2001*; *Yoganarasimha et al., 2006*; *Knight et al., 2011*; *Clark et al., 2012*), and on the geometric center of the platform to reset their phase. Both alternatives are consistent with the recent finding that grids in mice are destabilized in the dark, even when the HD signal is preserved (*Chen et al., 2016*; see also *Pérez-Escobar et al., 2016*).

Grids of similar spacing remain rigidly coupled during manipulations that elicit their global realignment or rescaling, including during hippocampal remapping (*Fyhn et al., 2007*; *Yoon et al., 2013*). By forcing the grids to dissociate from familiar cues, our experimental protocols offered additional opportunities for testing this rigidity. Simultaneously recorded grid cells responded consistently to the conflicting cues, even when their responses implied a large collective drift from the familiar reference frames. Furthermore, grid coordination was very high regardless of the extent of the animal's previous experience with the manipulation (*Figure 8—figure supplement 2*).

Grid coordination was found to be very high even for grids of markedly distinct scale (*Barry et al., 2007*; *Stensola et al., 2012*). This finding suggests the existence of internal mechanisms that can keep multi-scale grid populations geometrically coupled. Single-scale grid coupling is a built-in property of most attractor-network models of grid pattern generation (*Fuhs and Touretzky, 2006*; *McNaughton et al., 2006*; *Bonnevie et al., 2013*; *Burak and Fiete, 2009*). By construction, all grid cells within the same attractor network have identical scale and orientation while their phase differences are determined by the (fixed) network connectivity; thus, the grids remain geometrically coupled no matter their collective response to environmental modifications. However, coordination of grids of different spacing seems to require extending this model with an explicit mechanism for interlocking multiple networks operating at different spatial scales (*Knierim and Zhang, 2012*). Oscillatory models (*Burgess et al., 2007*; *Hasselmo et al., 2007*; *Blair et al., 2007*), on the other hand, do not rely on an extensive network in their basic form. Because they essentially work by turning velocity-modulated temporal oscillations into grid-like spatial oscillations, synchronizing the former might in principle 'synchronize' the latter, thus spatially interlocking the grids. Interlocked grids at different scales would require 'n:m' reciprocal entrainment, by which n cycles of one rhythm correspond to m cycles of the other, possibly at different phase-lags (*Zhang et al., 2009*; *Deshmukh et al., 2010*; *Belluscio et al., 2012*; *Brandon et al., 2013*). Such a framework, to our knowledge, has not been explored (but *Zilli and Hasselmo, 2010*; *Blair et al., 2014* investigated the synchronization of velocity-modulated oscillators for path-integration error-correction). In a third type of model, Hebbian learning of a grid-forming synaptic pattern is enabled by the spatiotemporal interaction of fast adaptive neural dynamics and spatial inputs that vary on a much slower behavioral timescale (*Kropff and Treves, 2008*; see also *Franzius et al., 2007*). The spacing of the grid depends on the time constant of the intrinsic adaptive dynamics. Common orientation in a population of grid cells and invariance of reciprocal phase relationships in multiple environments can be achieved in this model by plastic collateral connections at the time of learning (*Si et al., 2012*). Future work with this model could investigate how these geometric properties extend to similarly inter-connected populations of grid cells spanning multiple scales. Regardless of the class of models under consideration, the extensions required of existing models to enable multi-scale interlocking could produce theoretical insights into the functional reasons for the quantal organization of the grid spacing (*Barry et al., 2007*; *Stensola et al., 2012*; *Knierim and Zhang, 2012*; *Stensola and Moser, 2016*).

Further experimental and theoretical work is also needed to clarify the relationship between our observations and those by *Stensola et al. (2012)*. Those investigators observed a scale-dependent response of grid cells by compressing one axis of the enclosure: smaller-scale grids were 'chopped off' by the compressed apparatus, whereas larger-scale grids compressed along with it. Thus the mechanism responsible for the cross-scale grid coupling observed in our data may not always be active, and it is therefore unlikely to rely solely on genetically or developmentally hardwired networks. Alternatively, this mechanism may not apply to grid rescaling as it does to grid realignments, as anecdotally suggested by a mildly divergent rescaling followed by tight geometric coupling in a pilot protocol we described (*Figure 8—figure supplement 1*).

## Materials and methods

### Subjects and surgery

Eight male Long-Evans rats (Harlan Sprague Dawley Inc., Indianapolis, IN) were housed individually on a 12:12 hr light-dark cycle (dark cycle started at a consistent time point varying among rats between 10am–12pm). One rat was excluded from the analyses because its recordings did not yield any grid cells. Before the training protocol started, the rats were given one week for acclimation to

the facility and human handling. The rats were 5 ½ –6 ½ months old and 470–590 g at the time of surgery. Surgeries largely followed previous protocols and strategies for MEC recordings (*Hafting et al., 2005*; *Savelli et al., 2008*) to chronically implant a custom-built drive carrying 6–12 independently moveable tetrodes for electrophysiological recordings (rats 262, 263, 292: six tetrodes; rats 332, 334, 377, 387: 12 tetrodes). Under surgical anesthesia, a craniotomy was performed over the right hemisphere, the rostral edge of the transverse sinus was exposed, and the dura mater was removed from the adjacent brain surface. The drive was positioned so that the most posterior tetrode would penetrate the brain 250–470 μm anterior to the transverse sinus and 4.1–4.8 mm lateral to bregma. The remaining 5–11 tetrodes were spaced at ~300 μm intervals. The drive was oriented 5–10° anteriorly in the sagittal plane to increase the projected tetrode travel within MEC. Following recovery from surgery, 30 mg of tetracycline and 0.15 ml of a 22.7% solution of enrofloxacin antibiotic were administered orally to the animals each day. All animal care and housing procedures complied with National Institutes of Health guidelines and followed protocols approved by the Institutional Animal Care and Use Committee at Johns Hopkins University.

## Histology

Post-mortem histological analysis of recording locations followed standard procedures (e.g., *Savelli et al., 2008*). Briefly, rats were transcardially perfused with 4% formalin. The brain was extracted, stored in 30% sucrose formalin solution until fully submerged, and sectioned sagittally at 40 μm intervals. The sections were stained with 0.1% cresyl violet and used to identify tetrode tracks, based on the known tetrode bundle configuration. When the bundle contained >6 tetrodes, this procedure was aided by marker lesions produced by passing a positive 10 μA current for 10 s through the tip of 2 selected tetrodes >24 hr before perfusion. Finally, recording locations for each day were reconstructed along a tetrode track based on daily logs of tetrode movements.

## Apparatus and platform manipulations

The apparatus was located in a room (355 × 280 cm) with a constellation of visually prominent cues (*Figure 1*). A dim source of light was provided during the experiment by a ring-shaped lamp on the ceiling. Two types of square foraging platforms were used (137 × 137 cm). One had tall (36.5 cm) walls and paper floor that was replaced at the end of each session. The floor paper (light brown) was of a different color than the platform walls (dark brown). The other platform type had extremely short walls ('lips' <3 cm). In this platform, the color of the lips and floor was uniform, and the floor was not lined with replaceable paper. The floor was instead swept and then lightly mopped with water and dried between sessions, and extensively mopped with a 70% ethanol solution at the end of each day. Conspicuous urine or fecal matter was swiftly removed as soon as the rat moved away from it in either apparatus. No intentional geometric or local cues broke the symmetry of the square platforms. Each rat was exposed to only one of the two apparatus types: the high-wall type was used with rats 262, 263, 292, 332, and 334; the other type with rats 377, 387. All but one of the distal (room) cues (a small stool placed in one of the room corners) were visually accessible to rats foraging in both platforms (but the lower end of a few of them as well as the floors of the room were occluded in the platform with walls).

Both types of platforms were mounted on rails that allowed the rotations and translation of the platform with respect to the room. The experimental manipulations consisted of either rotations of the platform around its geometric center in the clockwise direction ('ROT20', 'ROT70', 'ROT30', 'ROT45') or the translation of the platform by half its size along one of its axes ('SHIFT'), starting from a standard ('STD') configuration of the platform. In the STD configuration the platform edges and room walls were parallel. The translation direction in SHIFT was parallel to the longer side of the room and kept constant throughout the experiments. Reproducibility of the manipulations across days and rats was aided by a laser pointer system that allowed precise placement of the platforms in repeated locations.

## Behavioral protocols

Training and experiments were started at or after the beginning of the circadian dark period. The animals were trained to forage in the platform in the STD configuration during a pre-surgery and post-surgery phase, before experiments were started. At the beginning of each training session the

rat was carried on a pedestal into one corner/quadrant of the platform, where it was released. During training this release location never changed. The experimenter paced and paused haphazardly around the room while throwing food pellets (chocolate sprinkles for some rats and 'bacon crumbles' [BioServ, NJ] for others) at a semi-regular rate and toward semi-random locations to encourage continuous locomotion and uniform sampling of the area and discourage stereotyped trajectories. On training and recording days the animals were kept at >80% (typically ~90%) of their free-feeding weight as necessary to motivate foraging behavior. The pre-surgery phase lasted 8–13 days and terminated when the animal foraged for >50 min with minimal interruption and relatively uniform spatial sampling. (This pre-training procedure led us to prioritize 'good runners' as surgery candidates.) After completion of the pre-training phase the animal was put back on a free-feeding diet for 4–16 days before surgery was performed and then again for 5–24 days during post-surgery recovery.

Experiments commenced when the rat's behavior was similar to its pre-surgery levels (after 6–10 days of post-surgery training) and well-isolated cells from putative MEC recording sites were encountered. The experiment day always began with an STD session, but the rat was released from a random corner/quadrant. In the following sessions, the platform was manipulated in full light conditions while the rat rested on its pedestal a few meters away, in sight of the whole experimental scene. No attempt at a disorienting the animal was ever performed. Lights were then dimmed again and the rat carried and released into a random corner/quadrant of the platform to start the new session. During electrophysiological recordings, the recording implant was connected to cables reaching a commutator mounted above the ceiling about the center of the ring source of light. The weight of these cables was counterbalanced by a pulley system. Each session (STD or manipulated) lasted 20–50 min to ensure ample and repeated sampling of the whole platform; 2–7 such sessions were performed each experiment day until scarce or no recording opportunities were estimated to remain (see below, up to 24 recording days). In some cases the STD session was repeated again during the day between manipulation sessions or at the end of the day.

## Behavioral and electrophysiological recordings

The animal position and head orientation were tracked via an array of multi-colored light-emitting diodes (LEDs) rigidly connected to the head implant, similar to previous experiments (*Savelli et al., 2008*). The LED signal was captured at 30 Hz by a CCD camera (JAI CV model 3300) through a small opening in the ceiling. Two identical cameras were present in the experimental room: one centered over the platform in the STD configuration and the other in the SHIFT configuration. The camera positions with respect to the platform in STD and SHIFT were as close as manual adjustment allowed, but some small offset was detected in the trajectory data and corrected as follows. The cumulative trajectories from all recording sessions in STD and SHIFT from a given rat were plotted and visually compared to determine a translation/rotation correction. Multiple rats were considered together in this procedure if they shared a cycle of experiments in which the camera settings and position had not been altered. These corrections were then applied to the trajectories from each session before using them in rate-map calculations (see below).

Tetrodes were made by twisting 12.5 or 17 μm nichrome wire (California Fine Wire Co., Grover Beach, CA). The tips were electroplated with gold, until the wire impedance was lowered to ~200 kOhms. The electrophysiological signal passed first through a unity-gain preamplifier headstage (Neuralynx, Bozeman, MT). For spikes, the signal was differentially amplified against a tetrode in a quiet area (usually layer I of MEC) and amplified between 2000 and 10,000 times and filtered in the 600–6000 Hz bandwidth. Waveforms crossing a >35 μV threshold on one of the four tetrode channels were sampled for 1 ms at 32 kHz on all four tetrode channels. These putative spikes were manually assigned to one or more putative cells (units) with the use of a custom spike-sorting program (WinClust; J. Knierim). Waveform characteristics (amplitude peak and area under waveform in most cases) on the four tetrode channels were plotted and cluster boundaries were manually drawn. For local field potential (LFP) recordings, the signal was differentially amplified 4000 times against the signal from a screw implanted in the skull, filtered in the 1–475 Hz bandwidth, and continuously sampled at 1 kHz.

Tetrodes were advanced by 40 μm up to a few hundred μm every day while the rat sat on the resting pedestal, until the tetrode tips were judged to be in MEC based on the detection of the theta rhythm in the LFP, unit activity, and anatomical considerations. One or two tetrodes in the array were moved faster in search of the phase reversal of the LFP theta wave known to occur near

the layer II/I boundary of MEC (*Alonso and García-Austt, 1987*), and were then left in a quiet site in layer I to act as references for spike recording on the other tetrodes. The distance traveled by these tetrodes were used to aid the estimation of anatomical position of all the other tetrodes in the array. Once the experiments started, tetrodes were usually advanced no more than 40 µm per day, except for tetrodes that were estimated to be still far from superficial MEC early on, and experiments continued until all useful tetrodes were judged to have reached layer I of MEC. We did not typically screen for grid cells as a precondition to perform experimental sessions on a given day, but we tended to run more sessions if previous days had yielded grid cells, or if the active observation of the ongoing experiment otherwise suggested the presence of grid cells.

## Rate maps

Rate maps were calculated in the camera reference frame from the tracked animal and spike positions. Epochs of immobility or very slow locomotion were excluded from these calculations. To estimate the rat's speed, its trajectory was first smoothed by convolving both raw x,y position time series with a clipped Gaussian mask with variance = 300 ms. The mask values were renormalized at each step. Rat's speed in cm/s was calculated for each of the 30 Hz frame intervals based on the rat displacement between the two consecutive frames. Epochs longer than 500 ms, in which the speed remained below 3 cm/s, were expunged from the original, unsmoothed trajectory and from the spike train. The rat positions left in the unsmoothed trajectory were then binned in 3 × 3 cm bins to produce an occupancy map of the rat's dwell time in each bin. The firing rate map was obtained by dividing the count of spikes occurring in each bin by the total occupancy in the same bin. Bins with less than 50 ms dwell time were marked as unoccupied (missing value) and excluded in visualization and analysis. The rate map was then smoothed by a clipped 2D Gaussian mask with 5 × 5 bins and variance = 2. The values of the mask were dynamically renormalized to account for unoccupied bins falling within the mask at each convolution step. If less than five occupied bins fell within the mask at any step, the output bin in the smoothed map was marked as unoccupied (missing value). 'Bootstrapped' rate maps were obtained by repeating the entire procedure after the spike train was resampled with replacement to obtain a new spike train of equal size 100 times, yielding 100 bootstrapped rate maps for each original rate map.

## Grid cell geometric parameters and inclusion criteria

Procedures for the evaluation of the grid structure ('gridness' test) and its geometric features were adapted from previous studies (*Hafting et al., 2005*; *Brandon et al., 2011*; *Stensola et al., 2012*) and applied to each rate map (i.e., experimental session) from a given unit independently. The crosscorrelogram of two rate maps was computed as the map of Pearson correlations of the two maps for all possible discrete displacements of the first map with respect to the second in the x and y directions (*Hafting et al., 2005*). The central bin of the crosscorrelogram represents the maps' correlation when no reciprocal displacement is applied. To reduce the occurrence of spurious correlations that result from small overlaps between the two maps at large displacements, the crosscorrelogram was populated only with correlations obtained from overlapping regions containing at least 100 bins.

The autocorrelogram is the special case of a crosscorrelogram calculated on two copies of the same rate map. The main geometric features of a grid were extracted from the autocorrelogram. Discrete regions of at least 20 contiguous bins of the autocorrelogram with correlation uniformly >0.1 were first identified as correlation fields. Three canonical grid axes running through the centers of mass of the correlation fields were chosen similarly to (*Stensola et al., 2012*): AX0 is the grid semi-axis that is closest in direction to the semi-positive abscissa; AX1 (AX2) is the first grid semi-axis encountered past AX0 in the CCW (CW) direction. The *orientation* of the grid is defined as the average direction of these canonical semi-axes. The *scale* of the grid is defined as the average distance of the three correlation fields (their centers of mass) defining the canonical axes from the center of the autocorrelogram, converted to cm according to the size of the rate map bins. *Elliptical distortion* is measured by an elliptical index (ranging from 0 to 1) defined as 1 - B/A, where B and A are respectively the length of the shorter and longer axis of the ellipse fit to the centers of mass of the six correlation fields most closely surrounding the central field.

Gridness scores were calculated similarly to prior papers (*Hafting et al., 2005*; *Brandon et al., 2011*). If the elliptical index was >0.05, the rate map was 'stretched' along the direction of the shorter axis so as to correct the distortion. The autocorrelogram, the seven most central correlation fields, and their centers of mass were then recomputed from this rate map. The annulus concentric with the autocorrelogram that contained the new six putative hexagon vertices was isolated from the rest of the autocorrelogram. The inner/outer radii defining this annulus were chosen as D ± 1.2 cR, where D is the average distance of the 6 centers of mass from the center of the autocorrelogram and cR is the estimated radius of the most central correlation field of the autocorrelogram. Pearson correlations between two rotationally offset copies of the annulus were computed. The gridness score is the minimum of the correlations obtained at rotational offset 30° and 90° minus the maximum obtained at 30°, 120°, and 150°.

In most previous studies (e.g., *Langston et al., 2010*; *Wills et al., 2010*; *Koenig et al., 2011*; *Brandon et al., 2011*), a threshold on the gridness score was used for grid cell classification. This threshold does not depend only on the analysis of the firing properties of the cell to which it is applied. Rather, it is a single value subjectively chosen by the investigator or statistically derived from the whole dataset (including non-grid cells; see discussion on shuffling below). Visual inspection of rate maps suggested to us that the exclusive use of a single gridness score threshold, however determined, could not keep the rate of both false positives and false negatives at a satisfactory level in our dataset and for our study's goals. Our analyses were particularly sensitive to the accuracy of the estimation of grid parameters, but we did not find the gridness score to provide a reliable measure of how 'clean' the grid was. The following individual criteria were therefore derived *ad hoc* and a rate map was classified as one produced by a grid cell if all criteria were met:

1. The gridness score was ≥0.1.
2. All six correlation fields defining the annulus could be identified as described above.
3. The angles subtended by the grid semi-axes were >30° and <90°.
4. The elliptical index of the autocorrelogram was <0.5.
5. The distance of the correlation fields from the ellipse was never greater than 20% of their distance from the center of the autocorrelogram.
6. The scale of the grid was <125 cm (putative larger grids could pass the test, but some of their vertices were almost entirely cut off the platform (137 cm x 137 cm), making their autocorrelogram-based geometric characterization ambiguous).
7. The gridness score was ≥0.1 for at least 95 out of the 100 bootstrapped rate maps when the procedure was repeated starting from these maps.

In the last step, we did not use the typical method of shuffling the spike train relative to the position time series to test for statistical significance of grid cells (e.g., *Langston et al., 2010*; *Wills et al., 2010*; *Boccara et al., 2010*), but instead used the bootstrapping of spike trains described above. The typical shuffling procedures destroy the spatial specificity of firing of the cell, and thus they are appropriate only to test whether a cell has significant spatial tuning, but not whether a spatially selective cell fires in a particular spatial pattern. The bootstrapping procedure, in contrast, does not destroy all spatial correlation in the firing. It is therefore especially useful to distinguish noisy but stable grids from noisy grids spuriously produced by fortuitous spatiotemporal fluctuations of the cell's firing rate. The gridness scores for the rate maps eventually accepted by this selection procedure were generally far greater than the 0.1 threshold used in steps 1 and 7 of the procedure (mean 1.26 ± 0.28 S.D., 5%ile = 0.68, 95%ile = 1.58). 3780 rate maps from 1332 units recorded from seven rats in distinct sessions of the same day were subjected to the selection procedure. 758 rate maps from 308 units passed the selection: seven units in rat 262, 46 units in rat 263, 14 units in rat 292, 39 units in rat 332, 52 units in rat 334, 99 units in rat 377, 51 units in rat 387. (Some of these units were recorded at anatomically close locations from the same tetrode on successive days and might correspond to the same cell—see *'thinning' of the dataset* below to mitigate related statistical concerns.) Because day to day decisions on the number of sessions to run and on electrode adjustment were influenced by knowledge of the presence of grid cells in the recordings, the proportions of cells that were found to be grid cells are prone to these sampling biases and are not intended to represent well-controlled, biological estimates of grid cell prevalence.

## Grid geometric response in different reference frames

To evaluate the geometric response of a grid cell to a given manipulation, we considered the rate maps of the grid cell in the STD and manipulation sessions if both rate maps satisfied the grid cell classification criteria. Rate maps were always calculated with respect to the camera reference frame, using one camera for all rotation manipulations (ROT20, ROT70, ROT30, ROT45), and a second camera identically centered with respect to the translated platform in the translation manipulation (SHIFT, which is dealt with separately, see below). The rotation performed by the grid was calculated as the difference of grid direction in the STD and manipulated conditions. If more than one STD session was run on the same day, the last STD run before the manipulated session under consideration was used as a reference for this session. Because of its 60° symmetry, rotations of an ideal hexagonal grid can be unambiguously defined within a circular range between −30° and 30°, where −30° and +30° are equivalent.

To calculate the grid rotation with respect to the platform reference frame, the rotation of the platform was subtracted from the measured grid rotation in the room (producing positive angles for over-rotations and negative angles for under-rotations of the grid with respect to the platform). The grid rotations thus computed, as well as the platform rotation when indicated on plots, are reported within the [−30°, 30°] grid angular range. For example, a 70° CW rotation of the platform is equivalent to a 10° CW rotation in grid angular space, in the sense that a grid that perfectly tracks the platform in ROT70 displays a change of direction of 10° between the STD and ROT70. Analogously, the platform rotation by 45° CW in ROT45 is equivalent to 15° CCW in grid angular space.

Similarly, we computed the grid rotation with respect to the reference frame given by the platform geometry (geometric reference frame). This geometric reference frame was specified by the 90° symmetric configuration of the square platform that implied the smallest rotation with respect to the room. The only manipulations in which the geometric reference frame dissociated from the platform reference frame (physical reference frame) were ROT70 and ROT45, for which the geometric reference frame rotated respectively 20° CCW and 15° CW in grid angular space. The rotation of the grid with respect to the geometric reference frame was calculated as for the platform reference frame: the rotation of this reference frame was subtracted from the grid rotation measured in the room and reported in the [−30°, 30°] grid angular range.

After the rotation of the grid was determined, we measured the grid phase shift that was required to complete the alignment of the grid in the STD and the manipulation sessions. These calculations were also dependent on the reference frame. To calculate the grid phase shift in the room reference frame in all types of manipulations (other than SHIFT), we rotated the STD rate map by the grid rotation measured in the room reference frame, so as to have the grids in the two conditions directionally aligned. We then computed the crosscorrelogram of the two rate maps. The grid phase shift was given by the vector representing the displacement of the center of mass of the most central correlation field in the crosscorrelogram from the center of the crosscorrelogram. The phase shift relative to the platform reference frame was analogously computed on a new crosscorrelogram obtained after the STD rate map was rotated by the same angle of rotation applied to the platform (e.g., the full 70° rotation for the platform frame in ROT70) ± the grid rotation in the platform reference frame (see above). An analogous procedure was also used to calculate the phase shift relative to the geometric reference frame, if this frame differed from the platform frame of reference. For example, a grid rotating 5° CW in the room reference frame in ROT70 performs a rotation of 5° CCW relative to the platform reference frame (because the expected rotation angle by a grid perfectly tracking the frame = 10° CW, due to the grid 60° symmetry) and a rotation of 25° CW relative to the geometric reference frame (because the expected rotation angle by a grid perfectly tracking the frame = 20° CCW). Accordingly, the phase shift for the platform frame is extracted from the crosscorrelogram of the STD and ROT70 rate maps after the STD rate map was rotated by 70° CW +5° CCW = 65° CW, whereas the phase shift for the geometric frame is extracted from the crosscorrelogram obtained after rotating STD rate map by 20° CCW +25° CW = 5° CW.

In SHIFT experiments, the geometric and platform reference frames coincide and are directionally aligned with the room reference frame. Grid rotation is therefore the same for all the reference frames. Grid phase shifts, however, will be different if calculated relative to the room or the platform/geometric frame. We first apply the same step as above: the STD rate map is rotated to equalize its grid direction with the SHIFT grid, and the phase shift is measured in the crosscorrelogram of

the two maps. Because both rate maps are calculated in the platform reference frame (from two cameras, see above), this is the grid phase shift relative to both the platform and geometric reference frames. To compute the phase shift relative to the room, we need to compare the observed platform-based phase shift with the one expected if the same grid were to remain perfectly anchored to the room and the platform were to reveal a different region of its pattern (i.e. corresponding to a null, room-based phase shift). The platform-based phase shift expected in this scenario occurs along the platform translation axis and equals the remainder of the division between the magnitude of the platform translation (68.5 cm) and the period of the grid projected along the translation axis. This phase shift can be expressed as a fraction of the grid period (projected on the translation axis, which is the x axis of the autocorrelogram) in radians:

$$\alpha = 2\,\pi\,\frac{68.5}{grid\ scale * \cos(grid\ orientation)}$$

after $\alpha$ is normalized between $-\pi$ and $\pi$. The observed platform-based phase shift can be analogously expressed as

$$\beta = 2\,\pi\,\frac{grid\ phase\ shift\ in\ platform}{grid\ scale * \cos(grid\ orientation)}$$

$\gamma = \beta - \alpha$ represents the phase shift seen in the room frame, also expressed in radians. After normalizing $\gamma$ between $-\pi$ and $\pi$, the room-based phase shift is calculated in cm as

$$room\ phase\ shift = \gamma * grid\ scale * \frac{\cos(grid\ orientation)}{2\,\pi}$$

## Geometric coupling of grids

We analyzed the geometric coordination of grid cells in any manipulation by two measures of grid coupling in pairs of simultaneously recorded grid cells. We wanted to compare coordination of grids both within similar scale groups and between different scale groups. However, a complication with the second measure (described below) arises when considering that grids of the same spatial scale can have high spatial correlations if their phases and orientations overlap, but grids at different spatial scales by definition cannot produce high correlations. Thus, for these analyses, we only analyzed grid pairs with quantitatively distinct spatial firing patterns (Pearson correlation of STD rate maps < 0.5) to minimize the potential influence of this asymmetry in possible correlations.

The first measure was the absolute difference between the rotations of the first grid and second grid with respect to their STD condition. Small values of this measure indicate that the two grids rotated by about the same angle, while larger values are indicative of a directional dissociation of the two grids caused by the manipulation. The second measure took into account the whole spatial distribution of firing expressed by the rate maps of the two grids. Its goal was to verify if a single, rigid transformation of both STD rate maps can produce firing patterns that are spatially correlated with those observed in the manipulated session, for both cells at once (a 'joint correlation'):

Let $RM1_{STD}$ and $RM2_{STD}$ be the rate maps of grid cells 1 and 2 in STD, and $RM1_M$ and $RM2_M$ their rate maps in the manipulated condition, with $RM1_{STD}$, $RM2_{STD}$, $RM1_M$, and $RM2_M$ calculated in a common reference frame:

1. Calculate the rotation magnitudes and directions of $RM1_{STD}$ and $RM2_{STD}$ relative to STD (using the methods described above) and calculate the average rotation $\rho$ of the two rate maps
2. Calculate the phase shift vectors for $RM1_{STD}$ and $RM2_{STD}$ relative to STD (using the methods described above) and calculate the average phase shift vector $\gamma$ of the two rate maps
3. Individually mean-center and normalize $RM1_{STD}$, $RM2_{STD}$, $RM1_M$, and $RM2_M$
4. Stack ($RM1_{STD}$, $RM2_{STD}$) into a 3D vector $V_{STD}$ by aligning the two rate maps along the Z axis
5. Produce an analogous 3D vector $V_M = (RM1_M, RM2_M)$ as in 4.
6. Rotate $V_{STD}$ by $\rho$ around the Z axis and shift it by $\gamma$ in the X,Y plane
7. Linearize vectors $V_{STD}$ and $V_M$ into 1D vectors so that corresponding entries in $V_{STD}$ and $V_M$ preserve the correspondence of rate maps and spatial bins.
8. The joint correlation is the cosine of the angle subtended by the linearized $V_{STD}$ and $V_M$

Control distributions for both measures of coupling (rotation difference and joint correlations) were obtained after the alignment of each grid pair was randomly perturbed 10 times. Each of the

STD rate maps was independently rotated by an angle drawn from a uniform distribution between −30° and 30°. Each of the rate maps in the manipulated conditions was shifted in a direction drawn from a uniform distribution between −180° and 180° and by a distance drawn uniformly between 0 and the scale of the grid.

## Statistics

In studies like this, measurements of functional properties of single units—presumed to represent different cells—are typically considered as the independent and identically distributed samples on which statistical testing is based. Accordingly, the experimenter should try to minimize the chances that the same cell was included more than once in the dataset submitted to a given test. We note that this approach might not be suitable for grid cells. It is not clear whether sampling from different, simultaneously recorded grid cells provide observations to be considered more statistically independent than sampling from the same grid cell in repeated experiments. In fact, grid cells of similar scale have been shown to act in a geometrically coordinated fashion (*Fyhn et al., 2007*; *Yoon et al., 2013*) possibly providing redundant sampling of the state of a single and functionally cohesive neural network. Grid cells of different scale have been shown to dissociate geometrically in certain experimental manipulations (*Stensola et al., 2012*), but they may also reflect coordinated action in other conditions, as described in the present study.

A comprehensive solution to this statistical problem is beyond the scope of the present study and we opted to keep with the conventional approach used in previous grid cell studies. Our primary analyses were carried out using each grid cell recorded as an independent data point. To reinforce the conclusions from the primary analyses, we report data from individual rats where appropriate. We also repeated each statistical test in a 'thinned' dataset, obtained by enforcing the constraint that no two units recorded in different days from the same tetrode can be both present in the dataset if their estimated anatomical distance along the tetrode trajectory was less than 150 μm. The constraint was generally enforced on samples intended as the combination of the unit itself (defined as rat, day, tetrode, and spike sorting cluster identifiers) and the manipulation type: a unit that is discarded for one manipulation may be retained for another, depending on whether the suspected duplicate from another day had been subjected to the same manipulation. The only exception in which we thinned by unit identity alone, with no consideration of the manipulation type, was the comparison of proportions of cells passing the gridness test in different manipulations for rat 387 (section "Neural correlates of the local VS. remote cue conflict"). The thinning procedure was completely automated with no manual intervention. In all but a few cases (specified in the text), all results and conclusions from the data were consistent between the full and the thinned data sets (although the p values of the tests were sometimes larger in the thinned datasets, but still statistically significant except where noted).

Because of the grid 60° symmetry, grid directions and rotations were defined in a circular range from −30° to +30° and wrapping around at these extremes (i.e. −30° =+30°). Accordingly, to perform statistical tests and comparisons on angles denoting grid directions and rotations, we employed circular statistical methods after the [−30°,+30°] range and sample angles were linearly mapped onto the full circle spanning the [−180°,+180°] range. Test statistics are reported in this transformed angular space, whereas descriptive statistics are reported in the original [−30°,+30°] range, unless otherwise noted. Rao's test for circular uniformity (*Zar, 1999*) was used to determine whether a set of data points was significantly clustered around a mean value. Tests for whether a set of circular data points was significantly different from a hypothesized value was performed by calculating the 95%, 99%, 99.9%, and 99.99% confidence intervals (C.I.) around the data set (*Zar, 1999*). The p value was designated corresponding to the largest confidence interval that excluded the hypothesized value (e.g., $p<0.0001$ if the value was outside the 99.99% C.I., $p<0.05$ if it was outside the 95% C.I., and n.s. if the value was within the 95% C.I.). For these statistical tests, we only used datasets with $n > 7$ (as prescribed in *Zar, 1999*). For noncircular data, statistical significance was tested with nonparametric statistics, as noted in the results section, except where indicated. All statistical comparisons were two-sided. Data are described as mean ± S.D., unless otherwise noted.

## Acknowledgements

We thank Geeta Rao for critical insights into apparatus design; Bill Nash and Bill Quinlan for apparatus construction; Yuna Shon, Amanda Smolinsky, and Geeta Rao for assistance with experiments; Sachin Deshmukh, Joseph Monaco, Cheng Wang, William Hockeimer, Kimberly Christian, for helpful comments on the manuscript; Hanne Stensola and May-Britt Moser for recommendations on surgery techniques. Data analysis and plotting in this study used the open-source software python/numpy/ scipy/matplotlib, as provided in the "Anaconda" distribution by Continuum Analytics, Austin, TX. Supported by International Human Frontier Science Program Organization Grant number LT00683/ 2006 C; NIH grants R01 NS039456 and R01 MH079511 from the U.S. Public Health Service.

## Additional information

### Funding

| Funder | Grant reference number | Author |
| --- | --- | --- |
| National Institute of Neurological Disorders and Stroke | R01 NS039456 | James J Knierim |
| Human Frontier Science Program | LT00683/2006-C | Francesco Savelli |
| National Institute of Mental Health | R01 MH079511 | James J Knierim |

The funders had no role in study design, data collection and interpretation, or the decision to submit the work for publication.

### Author contributions

FS, Designed experiments, Performed surgeries, Ran experiments, Performed histological analysis, Analyzed and interpreted electrophysiological recordings, Wrote manuscript; JDL, Ran experiments, Performed histological analysis, Commented on manuscript; JJK, Designed experiments, Performed histological analysis, Wrote manuscript

### Author ORCIDs

Francesco Savelli, http://orcid.org/0000-0002-8588-0865
James J Knierim, http://orcid.org/0000-0002-1796-2930

### Ethics

Animal experimentation: All animal care and housing procedures followed the recommendations in the Guide for the Care and Use of Laboratory Animals of the National Institutes of Health and protocols approved by the Institutional Animal Care and Use Committee at Johns Hopkins University (Protocols RA08A540 and RA11A486).

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
