## [Decision Letter]

Thank you for submitting your article "Framing of grid cells within and beyond navigation boundaries" for consideration by *eLife*. Your article has been favorably evaluated by Timothy Behrens (Senior Editor) and three reviewers, one of whom is a member of our Board of Reviewing Editors. The following individuals involved in review of your submission have agreed to reveal their identity: César Rennó-Costa (Reviewer #2); Alessandro Treves (Reviewer #3).

The reviewers have discussed the reviews with one another and the Reviewing Editor has drafted this decision to help you prepare a revised submission.

Summary:

The reviewers all felt that this paper was a significant advance in understanding the behavior of grid cells in the context of conflicting local and distal cues. They agreed that the study provides useful information for theory and experiment. The reviewers all felt that the methods were sound and experiments carefully executed.

Essential revisions:

A specific shared comment by the reviewers was that the text and figures should be clarified. They also felt that the Discussion should be focused and shortened.

*Reviewer #1:*

This paper looks at grid cell response changes in a context where a local platform moves or rotates in a large room with prominent landmarks. The authors show clearly that distal cues do indeed contribute to grid maps. The study systematically goes about showing this. They compare local vs. distal cues for rotation at different angles, and translation. They find that even for moderate platform movements, there is a small but consistent drag from distal cues. There is a consistent set of cases where ambiguous rotations are remapped according to the distal cues.

Overall, this is an interesting advance of the literature on place cell remapping and distal vs. local cues, to grid cells.

I would like the authors to further examine a point arising from their experiments on rat and manipulation-specific stability. Is the difference between rats just the sampling of cells, say from subsets of grid cells with potentially different cue responses? From Figure 8 and related text about the coordination between cells in an animal it looks like it is between rats, but I would like the authors to discuss.

It would be interesting if there were a behavioral correlate of differences between rats. For example, the authors discuss rat 387, which seemed to experience a strong influence of the room frame of reference. The authors clearly have full tracking data. Is there anything in the behavior that might correspond to this physiological measurement?

*Reviewer #2:*

The article provides evidence that rats can use both local cues (accessible through tact) and distal cues (reachable through sight) to anchor the allocentric reference frame of the internal map to the environment (grid cells). The main findings are: (1) one cue type is dominant; grid cells pattern will change coherently to one type or the other whenever the cues are independently modified; (2) modifications are not limited to rotations and are also observed for translations indicating that the underlying mechanism is not just a head-direction system reset; (3) the changes are animal- and modification-specific and remain constant across many days; (4) grid cells from different scales change coherently; and (5) conflictual cues might lead to grid degradation in specific cells of specific animals.

I do not find the reported results surprising. Similar results have been reported for place cells whereas grid cells realign following changes in the place cells firing patterns. However, the authors present countervailing evidence to the emerging view that local borders are the sole determinants of the grid cells firing pattern. Therefore, despite the lack of a "surprising factor", the work is timely and relevant. The paper provides many "missing pieces of the puzzle" of how grid cells actually work and I can see myself in the near future, as a theoretical scientist, using this manuscript as reference. Importantly, the methodology seems to be rigorous, aligned to the community standards and is scientifically sound. Thus, I would recommend its acceptance for publication granted that my concerns (described below) and minor clarity issues in the manuscript are solved.

The authors base their analysis on unusually long exploratory sessions (20-50 minutes, compared to the usual 10-20 minutes in free exploratory sessions) reflecting the authors' attention to the statistics. However, it concerns me whether drift in the grid cell activity during such a long period contaminates the statistics. Mainly, the grid cell pattern could possibly switch between different anchors in time. If so, it would interfere in the gridness values and grid orientation. To better understand this issue, the authors could examine the stability of the grid cells comparing the first and second halves of the exploration or come with a metric about how long does it take for the grid map to stabilize. They can compare these analyses in STD and conflictual conditions.

Some interesting work have shown that grid cells are not uniform across the environment and that large environments can get compartmented. The size of the platform in this study is rather large (137x137cm if compared to 80x80cm in other studies). I'm concerned if whether the reported dominance is uniform across the whole platform. One possible side effect of the compartmentation is that grid cells might not be regarded as grid cells (low grid score). The authors could segment the apparatus and analyze different parts separately. They can look whether the anchor in one side of the arena is the same as in another side (same for center vs. borders).

*Reviewer #3:*

This is an important contribution, reporting the effects on grid cell firing patterns of incongruent proximal and distal cues. The experimental work addresses issues I was (and likely many others were) wondering about, and in my view it should be published without delay. The analysis stays mostly close to the data, which I appreciate.

My only mild criticism has to do with the hypothesis testing formatting of some of the analyses, which is not necessarily the authors' choice as it may have been largely an attempt to meet presumed reviewer requests, and with the lengthy discussion, which by morphing it into almost a review paper blunts the impact of the experimental findings.

Hypothesis testing is only really appropriate if there is the expectation that one cause-effect relationship or at least one correlation should dominate all others, an expectation that can then be tested statistically. It is inappropriate in most neural coding contexts, where cause-effect relations are merely convenient short-hands for directed interactions, and interactions tend to be multifarious. It is particularly inappropriate in studies like this one, where competing interactions are quantitatively set against each other. In my view the authors should recast their analyses in the language of quantifying the strength of each interaction they look at, rather than extracting p-values. For example, in the three-way competition thought to occur in the ROT70 condition, would a low p for the "intermediate" hypothesis of room control be distinguishable for a specific combination of the two extremal hypotheses of platform and geometric control, with little room control?

Further, the emphasis on hypothesis testing at the single cell level, over the first 6 figures, largely obscures the splendid population result reported, perhaps not so perspicuously, in Figures 7, 8, which in my view should be highlighted. Panel 7C seems like an extremely strong result, which goes unnoticed in a crowded figure. Panel 7D might be equally important, but I cannot make out the pink and blue dots, and I am not sure what it says. Overall, Figure 7 points out how perfect gridness is a misconceived notion, to be handled with care, not used as a basis for all analyses. Figure.8 casts similar doubts on the notion of segregated grid modules.

In the Discussion, much space is devoted to a minute review of classical literature on cue control, whereas I would focus on grids, and in relation to grids on the 3 outstanding questions which these beautiful experiments pose:i) What makes a multi-peak cell appear, in an idealized setting, as a quasi-perfect triangular grid? I lean of course towards the proposal outlined in our adaptation model (Kropff and Treves, 2008), whereas the authors discuss only the rather implausible attractor and oscillatory interference models, which are inconsistent, in their pristine form, with a host of earlier data;ii) Can multiple grid maps coexist in the same module, in contrast to the Fyhn et al. 2007 result? This I think is the core suggestion one gets from Figure 7, and if validated it would open up an entirely new perspective on grid coding (entirely consistent, full disclosure, with our model calculations predicting a large storage capacity for incongruent grid patterns expressed by the same cells);iii) Are the modules segregated ab initio, e.g. in terms of connectivity, or do they self-organize and maintain a complex web of mutual interaction (again consistent, full disclosure, with our expectations from modelling work)?

---

## [Author Response]

*Essential revisions:*

*A specific shared comment by the reviewers was that the text and figures should be clarified. They also felt that the Discussion should be focused and shortened.*

*Reviewer #1:*

*[…] I would like the authors to further examine a point arising from their experiments on rat and manipulation-specific stability. Is the difference between rats just the sampling of cells, say from subsets of grid cells with potentially different cue responses? From Figure 8 and related text about the coordination between cells in an animal it looks like it is between rats, but I would like the authors to discuss.*

As the reviewer points out, the differences appear to be mostly rat-based and unlikely to emerge from the sampling of functionally distinct subsets of grid cells in different rats. This is an important point that we failed to emphasize appropriately. To address it we have added a table to the paper (Table 1), which is introduced as follows (see also Table 1 legend):

“The difference of response expressed by the bimodal distribution of rotations in ROT70 (Figure 3A) is unlikely to result from the sampling of functionally differentiated neuronal networks (Table 1).”

This table is relevant to the manipulation in which the differences between rats were most prominent, namely ROT70 and its bimodal rotation response distribution. The table describes the composition of each mode in terms of brain areas, rats, number of units (and number of tetrodes from which the units were recorded). A diverse set of anatomical regions is represented in both modes, whereas rat identity is almost perfectly segregated by mode, with the exception of 1 unit from rat 292 belonging to the left mode and 6 to the right mode. These 7 units were recorded in the same area (putatively parasubiculum) and by the same tetrode.

Moreover, as observed by the reviewer, Figure 8 on grid coordination also indirectly suggests that the response is unlikely to depend on sampling biases. To reinforce this indication, we have added information in the Results section on the number of pairs in the different clusters of Figure 8 that comprised grid cells recorded on different tetrodes and specifically from different areas:

“In most pairs the two grid cells were anatomically adjacent as they were recorded from the same tetrode, but in many other cases they were recorded on different tetrodes (SR1-3: 119, 48, and all 5 pairs, respectively). In some of the latter cases the recording sites were ascertained to be in different layers of MEC or in MEC and parasubiculum (46 in SR1 and 9 in SR2 by a highly conservative histological evaluation).”

We suspect that individual experience and/or apparatus type might be the factors influencing the response type, but statistical validation of this impression would require a far greater number of rats – hence we originally refrained from discussing these factors. We have briefly acknowledged these considerations at the end of the second paragraph of the Discussion section:

“In our experiments differences in the relative dominance of distal and local cues appeared to depend on individual differences between rats and/or the type of apparatus, rather than on the anatomical regions where the grid cells were recorded (Table 1, Figure 8). However, the statistical validation of this impression would require data from many more rats than considered in our study.”

*It would be interesting if there were a behavioral correlate of differences between rats. For example, the authors discuss rat 387, which seemed to experience a strong influence of the room frame of reference. The authors clearly have full tracking data. Is there anything in the behavior that might correspond to this physiological measurement?*

This is a compelling line of inquiry that we did not pursue explicitly in this study. Some basic insights into animal behavior are indirectly offered by our original attempts to mitigate concerns that grid distortions were a spurious effect of a behavioral change affecting rate map calculations. We briefly recall these attempts and their implications. Epochs of animal immobility – when the hippocampal spatial representation can stop tracking the animal position while “replaying” previously experienced trajectories in the present or other familiar environments – were not used in the calculation of the rate map, thus any distortion in the grids cannot be attributed to this phenomenon. The experimenter generally waited for the rat to cover the area multiple times in all type of sessions, rate maps were normalized by occupancy, and we reported high correlations of rate maps derived from the first and second halves of the session (median Pearson correlations > 0.66; this analysis has now been extended to all other rats in response to a comment from reviewer 2, see subsection “Grid coordination within and between scales”, first paragraph). These considerations seem to rule out pronounced differences in the coverage patterns across or within sessions. The loss of grid regularity but not of the multi-peaked nature of the rate maps seems to rule out the possibility that the representation of place fields in the rate maps was somehow compromised by behavioral aberrations.

In consideration of the reviewer’s question, we have more carefully examined rats’ velocities. The speed distribution in rat 387 is minimally higher in ROT70 than in the other types of manipulated sessions pooled together (median speed 12.7 vs. 12.4 cm/s), but similarly minimal differences were found in other rats too, or when other types of manipulations were singled out. In most cases, the sign of these differences did not depend on the size of the sliding time window used to compute the velocities (66ms and 1s) and the differences were found to be statistically significant (as assessed by Mann-Whitney or Kolmogorov-Smirnov tests on very large data sets pooled from all days). We refrain from interpreting these particular results, as the effect sizes are very small. As mentioned in the response to the previous comment, our impression is that the type of apparatus and individual differences played a role in the emergence of different responses in different rats, but the statistical validation of this impression requires a larger number of rats. By the same token, we think that a systematic and well controlled investigation of the link between individual behaviors and the relative influence of distal vs. proximal cues on grid cells may be prohibitively long with current recording strategies/techniques.

*Reviewer #2:*

*[…] The authors base their analysis on unusually long exploratory sessions (20-50 minutes, compared to the usual 10-20 minutes in free exploratory sessions) reflecting the authors' attention to the statistics. However, it concerns me whether drift in the grid cell activity during such a long period contaminates the statistics. Mainly, the grid cell pattern could possibly switch between different anchors in time. If so, it would interfere in the gridness values and grid orientation. To better understand this issue, the authors could examine the stability of the grid cells comparing the first and second halves of the exploration or come with a metric about how long does it take for the grid map to stabilize. They can compare these analyses in STD and conflictual conditions.*

As the reviewer points out, the unusually longer sessions were motivated by sampling/statistical objectives. The experimenter typically waited until the rat roughly covered the whole arena >2 times, so that estimates of local firing rates could be based on repeated visits of the same spot. The concern raised by the reviewer is important. To address it we calculated the Pearson correlation of the two rate maps derived respectively from the first and second half of each session that lasted longer than 30 minutes in all rats but rat 387, which had already been separately analyzed in this way. These correlations were found to be very high in both STD and manipulated sessions, suggesting the absence of major intra-session grid drift or change of anchoring. The correlations in STD and the manipulated sessions were not found to be significantly different. We added this information at the end of the subsection “Neural correlates of the local vs. remote cue conflict”:

“To mitigate concerns that these idiosyncratic responses, as well as the previously described minor and major dissociations from the platform reference frame, resulted from spatially unstable grids, we extended the analysis employed above for rat 387 to the recording sessions obtained from the other rats. […] These correlations were found to be very high in both STD (median correlation > 0.68 in each rat) and manipulated sessions (median correlation > 0.69 in each rat), suggesting that there was no major intra-session grid drift or change of anchoring within a session (correlations in STD were not different from those in manipulated sessions: Mann-Whitney test, U = 32572, p > 0.48, all rats pooled together).”

*Some interesting work have shown that grid cells are not uniform across the environment and that large environments can get compartmented. The size of the platform in this study is rather large (137x137cm if compared to 80x80cm in other studies). I'm concerned if whether the reported dominance is uniform across the whole platform. One possible side effect of the compartmentation is that grid cells might not be regarded as grid cells (low grid score). The authors could segment the apparatus and analyze different parts separately. They can look whether the anchor in one side of the arena is the same as in another side (same for center vs. borders).*

These are very important points. Local distortions certainly occur, not limited to the selective loss of gridness in ROT70 concerning rat 387. This was clear to us from close inspection of the rate maps. We are very interested in knowing how these local distortions relate to the recent work the reviewer hinted at (mainly Stensola et al. 2016 and Krupic et al. 2016). We had noticed that some of these distortions (including 387) might result from an anisotropic balance of relative dominance of the two reference frames, possibly center vs. periphery, as the reviewer also wondered. Compartment-based analyses of the kind employed in the cited studies and recommended by the reviewer can in principle quantify these effects. But for compartmentalized analyses to work, enough vertices of the grid would need to fall within each compartment to make a reliable local autocorrelogram. Unfortunately, we anticipate that this would not be the case for most of our data. Prior to our original submission, we made progress on devising alternative analyses, but we realized that such effort would evolve into a study of its own that is best dealt with separately.

To gauge the extent to which the issue raised by the reviewer affects our dataset, we identified all the manipulated sessions that failed to pass our gridness test even if they did so in the previous and next session (26 sessions from 3 rats; we excluded rat 387, in which similar analyses were already described in the manuscript). This criterion minimizes the chance that the loss of gridness was caused by mounting noise in the recordings, so we reasoned that these were the most likely candidates in which a loss of gridness caused by a compartmented response could be observed. Visual inspection of their rate and spike/trajectory maps suggested that in the majority of these cases the failure to pass the gridness test could be likely attributed to (1) poorer spatial sampling (i.e. not uniformly repeated multiple times) that did not pass the bootstrap part of our gridness test or (2) a large grid dissociating from the platform and losing a vertex (or a large part of it), making its grid structure no longer detectable. In no case did the number of visible vertices seem to allow the type of analysis discussed above. Because we are not able to adequately address the reviewer’s concern with our current data, we believe it would be best to not discuss the issue in the paper. However, if the reviewer insists that this is an important point that we must address, we will add a discussion after consultation with the editors.

*Reviewer #3:*

*[…] My only mild criticism has to do with the hypothesis testing formatting of some of the analyses, which is not necessarily the authors' choice as it may have been largely an attempt to meet presumed reviewer requests, and with the lengthy discussion, which by morphing it into almost a review paper blunts the impact of the experimental findings.*

*Hypothesis testing is only really appropriate if there is the expectation that one cause-effect relationship or at least one correlation should dominate all others, an expectation that can then be tested statistically. It is inappropriate in most neural coding contexts, where cause-effect relations are merely convenient short-hands for directed interactions, and interactions tend to be multifarious.*

We are generally sympathetic to these comments. One of us (F.S.) personally harbors strong reservations on frequentist statistics and “classical” hypothesis testing in general and favors a Bayesian perspective on data analysis and inference. (We note that in the Methods section we originally discussed some challenges to statistical assumptions that are common in this type of study, albeit of a different kind than here pointed out by the reviewer.)

*It is particularly inappropriate in studies like this one, where competing interactions are quantitatively set against each other. In my view the authors should recast their analyses in the language of quantifying the strength of each interaction they look at, rather than extracting p-values. For example, in the three-way competition thought to occur in the ROT70 condition, would a low p for the "intermediate" hypothesis of room control be distinguishable for a specific combination of the two extremal hypotheses of platform and geometric control, with little room control?*

In most cases those tests were not employed to establish which reference frame dominated the representation, but were used to validate the grid geometrical dissociation from each given reference frame (including what looked to be the dominant one). The only exception is the comparison of phase shift magnitude in the last paragraph of the subsection “Grid anchoring to both local and remote cues in rotation experiments”. Our extensive use of scatterplots to display the entire distribution of grid rotation and phase shift in almost any circumstance (especially through Figure 5—figure supplements 1-7) reflects our impression that the strength of interactions is amenable to direct visual assessment. Dominance of a given reference frame (or lack thereof) should be immediately evident by visual inspection of the whole dataset.

In spite of our general agreement with the reviewer on these epistemological issues, we do not share the reviewer’s opinion that discarding the statistics would be beneficial to the dissemination of our findings. *eLife* has stringent requirements in terms of data and statistics presentation, in line with the standards that the general biomedical audience has come to expect. Another reviewer praised our attention to statistics. Because of journal requirements and current standards in the field, we have chosen to keep the statistics as they are.

*Further, the emphasis on hypothesis testing at the single cell level, over the first 6 figures, largely obscures the splendid population result reported, perhaps not so perspicuously, in Figure 7, 8, which in my view should be highlighted. Panel 7C seems like an extremely strong result, which goes unnoticed in a crowded figure. Panel 7D might be equally important, but I cannot make out the pink and blue dots, and I am not sure what it says. Overall, Figure 7 points out how perfect gridness is a misconceived notion, to be handled with care, not used as a basis for all analyses. Figure 8 casts similar doubts on the notion of segregated grid modules.*

Thank you. We have split Figure 7 (also in response to comments by reviewer 2 above). Panels E and F were moved to a different figure (current Figure 7). Panels A-D are now in Figure 6 by themselves and were enlarged to address the issue of their clarity. We also replotted Figure 7D (now Figure 6D) without transparency (α = 0). We originally used a high level for transparency to render overlapping dots but we find that the new version has more color contrast without having lost information.

*In the Discussion, much space is devoted to a minute review of classical literature on cue control, whereas I would focus on grids, and in relation to grids on the 3 outstanding questions which these beautiful experiments pose:i) What makes a multi-peak cell appear, in an idealized setting, as a quasi-perfect triangular grid? I lean of course towards the proposal outlined in our adaptation model (Kropff and Treves, 2008), whereas the authors discuss only the rather implausible attractor and oscillatory interference models, which are inconsistent, in their pristine form, with a host of earlier data;ii) Can multiple grid maps coexist in the same module, in contrast to the Fyhn et al. 2007 result? This I think is the core suggestion one gets from* Figure 7*, and if validated it would open up an entirely new perspective on grid coding (entirely consistent, full disclosure, with our model calculations predicting a large storage capacity for incongruent grid patterns expressed by the same cells);iii) Are the modules segregated ab initio, e.g. in terms of connectivity, or do they self-organize and maintain a complex web of mutual interaction (again consistent, full disclosure, with our expectations from modelling work)?*

We have now removed much of the detailed review on cue control (extensive pruning throughout the Discussion section). We also share the reviewer’s captivation with these three theoretical points. Some of the data we presented intriguingly speak to these points, along the lines suggested by the reviewer. However, with regard to (i) and (ii), the data were quantified in only one rat or another. Hence, we hesitate to speculate at length about their theoretical implications in this paper, but we address these points as follows:

With regard to (i) we more explicitly addressed the differences between grid distortions in rat 387 and previously documented cases of grid distortions in the Discussion:

“This highly selective grid disruption was triggered by the conflict between proximal and distal cues in the absence of any structural alteration of the platform. Thus, the grid disruption cannot be attributed to tensions induced by the platform frame alone (Barry et al., 2007; Derdikman et al., 2009; Stensola et al., 2012; Krupic et al., 2015) but probably to the grid system’s attempt to reconcile proximal and distal inputs.”

We also completed the discussion of the implications of multi-scale coordination by explicitly addressing the model of grid formation developed in Kropff and Treves 2008 and further extended in Si, Kropff and Treves 2012, alongside a slightly less detailed discussion of the attractor and oscillatory models of grid formation:

“In a third type of model, Hebbian learning of a grid-forming synaptic pattern is enabled by the spatiotemporal interaction of fast adaptive neural dynamics and spatial inputs that vary on a much slower behavioral timescale (Kropff and Treves, 2008; see also Franzius et al., 2007). […] Future work with this model could investigate how these geometric properties extend to similarly inter-connected populations of grid cells spanning multiple scales.”

With regard to (ii) we commented on the examples in Figure 7 (different grid maps in STD vs STD2) in the context of place cell global remapping and memory:

“A seeming counter-example to the involvement of grid framing in proper context recall was instead noticed in one rat in which different grid maps were produced in identical cue configurations at the beginning and end of the experimental day (STD vs. STD2, Figure 6B, Figure 6—figure supplement 2). But even in this case a short-term memory process appeared at play, since we anecdotally found the later grid map to reflect the recent history of experimental interventions (Gupta et al., 2014), which could be regarded as a form of contextual discrimination.”

With regard to (iii) we briefly pointed out the implications of the findings in Figure 8 for the nature and development of intra-module functional relationships:

“Thus the mechanism responsible for the cross-scale grid coupling observed in our data may not always be active, and it is therefore unlikely to rely solely on genetically or developmentally hardwired networks.”